# CALIBRATION ATTACK: A FRAMEWORK FOR ADVERSARIAL ATTACKS TARGETING CALIBRATION

## ABSTRACT

We introduce a new framework of adversarial attacks, named *calibration attacks*, in which the attacks are generated and organized to trap victim models to be miscalibrated without altering their original accuracy, hence seriously endangering the trustworthiness of the models and any decision-making based on their confidence scores. Specifically, we identify four novel forms of calibration attacks: *underconfidence attacks*, *overconfidence attacks*, *maximum miscalibration attacks*, and *random confidence attacks*, in both the black-box and white-box setups. We then test these new attacks on typical victim models with comprehensive datasets, demonstrating that even with a relatively low number of queries, the attacks can create significant calibration mistakes. We further provide detailed analyses to understand different aspects of calibration attacks. Building on that, we investigate the effectiveness of widely used adversarial defences and calibration methods against these types of attacks, which then inspires us to devise two novel defences against such calibration attacks.

## 1 INTRODUCTION

Adversarial attacks (Ren et al., 2020) have proven to be a critical tool to reveal the susceptibility of deep neural networks (Ibitoye et al., 2019; Zimmermann et al., 2022; Xiao et al., 2023), where, in a typical setup, adversarial examples are generated by introducing imperceptible perturbations on an original input to cause model misclassification. The existing attacks have mainly focused on trapping victim models to make incorrect predictions.

In this paper, we highlight a serious set of threats, named *Calibration Attack*, which focus on attacking the calibration of victim models by deceiving the models to be over or under-confident without modifying their prediction accuracy, hence endangering any decision-making based on a classifier's confidence scores. We briefly illustrate this in Figure 1.

Calibration attacks are insidious and arguably harder to detect than standard attacks, which we empirically demonstrate in Table 2, when using popular adversarial attack detection techniques. The harm presented from these attacks is that on the surface the model appears to still be making mostly correct decisions, but the level of miscalibration could make the model's decisions unusable for downstream decision making. Consider the case of autonomous driving, where after a set of detected street signs images are calibration-attacked, it can appear as though an object recognition model is performing well even when scrutinized, but the confidence levels on ambiguous cases that need review by further backup processing systems will be altered, and hence compromise the affected decision-making leading to potential complications.

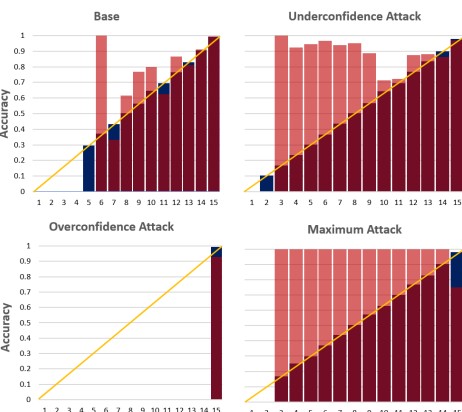

Figure 1: Calibration diagrams of a normal ResNet-50 classifier before and after three forms of *calibration attacks*. Red bars show the average accuracy on the test data binned by confidence scores and blue bars represent the actual average confidence in each bin. The yellow line depicts perfect calibration. Despite the accuracy being unchanged, the miscalibration is severe after the attacks.

In our study, we propose four forms of calibration attacks: *underconfidence attack*, *overconfidence attack*, *maximum miscalibration attack*, and *random confidence attack*. We test these newly identified attacks primarily in the black-box setup due to it being a more realistic and applicable threat scenario, but we also test white-box variations. We use benchmark image classification datasets to evaluate the models. We also analyze the adequecy of existing adversarial defenses and calibration techniques against maintaining good calibration post-attack, as well as formulate and study two novel forms of defence, Calibration Attack Adversarial Training (CAAT) and Compression Scaling (CS), helping to bring additional insights for how to best resist these calibration attacks.

To the best of our knowledge, this is the first study that provides a comprehensive investigation on calibration attacks, especially on the threat of inducing overconfidence. In summary, our main contributions are as follows. 1) We introduce and investigate a new form of adversarial attack, the *calibration attack*, and advocate that particular caution should be seriously taken for such attacks. 2) We construct four specific forms of calibration attacks and show their danger to typical deep learning models. Detailed analyses are provided on the effectiveness of different attacks and the vulnerability of different architectures of victim models. 3) We show the severity of calibration attacks on many models using current calibration and defence methods, and demonstrate the compromise often presented between robustness and calibration, but offer additional insights for future defences based on our two novel methods.

## 2 RELATED WORK

**Calibration of Machine Learning Models.** Calibration methods are generally divided into two types. Post calibration can be applied directly to the predictions of fully trained models at test time, which include classical approaches like temperature scaling (Guo et al., 2017), Platt scaling (Platt, 1999), isotonic regression (Zadrozny & Elkan, 2002), and histogram binning (Zadrozny & Elkan, 2001). Training-based approaches, however, typically add a bias during training to help a model to be better calibrated (Zhang et al., 2018; Thulasidasan et al., 2019; Kumar et al., 2018; Tomani & Buettner, 2021). In this work we investigate how a diverse range of calibration methods cope against calibration attacks and study the limitations that require them to be overhauled to deal with attacks, including the vulnerability of models on convolutional (Guo et al., 2017; Minderer et al., 2021), and Transformer architecture ViT (Dosovitskiy et al., 2021).

**Adversarial Attacks and Training.** We provide a more detailed overview of adversarial attack methods in Appendix A. Black-box attacks (Carlini & Wagner, 2017) have less information about victim models. Many methods of this type are based on some form of gradient estimation through querying the model and finding the finite differences (Bhagoji et al., 2018). In contrast, white-box attacks often have access to the full details of a victim model such as model architectures and gradients. For example, a typical example of this type, among many others, is the Fast Gradient Sign Method (FGSM) (Goodfellow et al., 2015).

In terms of adversarial training, many approaches have also been introduced (Stutz et al., 2020; Chen et al., 2022; Qin et al., 2021; Patel et al., 2021; Dhillon et al., 2018). For example, Stutz et al. (2020) improve the robustness of the models against previously unseen attacks by rejecting low-confidence adversarial examples through confidence-thresholding; Chen et al. (2022) design a post processing adversarial defence method against score-based query attacks. Nevertheless, none of the prior works comprehensively study calibration attacks and systematically investigate how well victim models would remain well calibrated under such attacks.

## 3 CALIBRATION ATTACK

In a standard multi-class classification task, we have a set of input vectors $\mathcal{X} = \{\boldsymbol{x}_1, \ldots, \boldsymbol{x}_N\}$ with dimensionality $\boldsymbol{d}$. True labels are denoted as $\mathcal{Y} = \{\boldsymbol{y}_1, \ldots, \boldsymbol{y}_N\}$, where $\boldsymbol{y}_i$ is a class label in a finite set $\{1, \ldots, \boldsymbol{K}\}$. The goal in neural network-based classification is to find a mapping to predict the class label of a given $\boldsymbol{x}$, i.e., $\mathcal{F} : \boldsymbol{x} \rightarrow (\hat{\boldsymbol{y}}, \hat{\boldsymbol{p}})$, where $\hat{\boldsymbol{y}}$ represents the predicted class label obtained by taking the argmax of the probability distribution $\hat{\boldsymbol{p}}$. This probability distribution $\hat{\boldsymbol{p}}$ is often generated using a softmax function on the output layer of the network, ensuring that $\sum_{i=1}^{K} \hat{\boldsymbol{p}}_i = 1$. The highest probability in $\hat{\boldsymbol{p}}$ corresponds to the predicted class and serves as the prediction confidence.

### 3.1 Objective of Calibration Attack

Calibration attack aims to generate adversarial samples to maximize the miscalibration function $M(\tilde{x}, k)$ for a predicted class $k$ and an adversarial sample $\tilde{x}$. As will be detailed in Section 3.2, we propose four specific forms of calibration attacks, in which the miscalibration function $M(\tilde{x}, k)$ has different implementations. The adversarial sample $\tilde{x}$ is created by adding adversarial noise $\delta$ to an input vector $x$: $\tilde{x} = x + \delta$. To ensure that the adversarial example and the original do not differ a notable amount and following convention, a maximum $l_p$-ball norm difference bounded by an $\epsilon$ is defined between them. For completeness of the paper, we include the details of adversarial constraint to be satisfied for certain $p \in \{0, 1, .., \infty\}$ norm:

$$\|\tilde{x} - x\|_p < \epsilon, \ s.t \ \hat{y} \neq y, \tag{1}$$

where $\epsilon$ controls the amount of perturbation allowed.

In general, our attacks are based on the most popular and easiest-to-measure view of calibration called classwise calibration that focuses on the predicted class (Kull et al., 2019). The equation below details the notion for all input datapoints $(x_n, y_n) \in \mathcal{D} = \{(x_n, y_n)\}_{n=1}^{N}$ in a dataset $\mathcal{D}$:

$$\mathbb{P}(y_n = \hat{y}(x_n) \mid \hat{p}_k(x_n) = p_k) = p_k, \tag{2}$$

where $p_k$ is the confidence score for the predicted class $k$. Any mismatch between the left and right hand sides of the equation creates undesirable miscalibration.

### 3.2 Four Types of Calibration Attack

Within the aforementioned framework, we propose to construct four forms of calibration attack: *underconfidence attack*, *overconfidence attack*, *maximum miscalibration attack*, and *random confidence attack*. The first two types serve as the basis for the latter, while the maximum miscalibration and random confidence attacks are developed to create a higher and more problematic degree of miscalibration compared to the base attacks.

**Underconfidence and Overconfidence Attack.** These two base attacks aim to solve the constrained optimization problem involving the miscalibration function $M(\tilde{x}, k)$ for the predicted class $k$. Specifically for the underconfidence attacks, the loss function is defined as:

$$M(\tilde{x}, k) = \hat{p}_k(\tilde{x}) - \max_{j \neq k} \hat{p}(\tilde{x}). \tag{3}$$

The aim is to reduce the confidence scores on all predictions. For overconfidence attacks, the loss function is $M(\tilde{x}, k) = 1 - \hat{p}_k(\tilde{x})$, and the objective is updated so that adversarial examples are crafted to have the highest level of confidence. Both of these come with the added constraint that the examples are generated to try to keep predicted class label $\hat{y}(x) = \hat{y}(\tilde{x})$, irrespective of true label $y$, and with the goal of ensuring that the maximum amount of calibration error over the set of adversarial inputs $\tilde{\mathcal{X}}$, $\max_{\tilde{\mathcal{X}}}(\mathbb{P}(y_n = \hat{y}(\tilde{x}_n) \mid \hat{p}_k(\tilde{x}_n) = p_k) - p_k)$ is produced.

The overall algorithm for the underconfidence and overconfidence attacks can be seen in Algorithm 1. The basis of our primary implementation of calibration attacks is the popular and highly effective Square Attack (SA) (Andriushchenko et al., 2020), a black-box score-based adversarial attack approach. It has attained state-of-the-art results in terms of query efficiency and success rate, even outperforming some white-box methods. Unlike the standard adversarial attack settings, we do not only run the attacks on correctly classified examples, but also the originally misclassified inputs, since they can also be further miscalibrated; attacking misclassified inputs in such a manner could be an additional avenue for adversaries to pursue to harm downstream decision-making. There are two basic forms of SA attacks, based on the $l_\infty$ or $l_2$ norm. We create a version of calibration attack for each norm. In addition to SA, we also create white-box variations of calibration attack using the effective Projected Gradient Descent (PGD) framework in the $l_\infty$ threat model. These attacks follow the same algorithm but have additional implementation details, which can be seen in Appendix B.

**Maximum Miscalibration Attack.** The main principle behind the maximum miscalibration attack is that to create the most amount of miscalibration in classifier for a given set of data, one needs to have all of the incorrectly classified samples have near 100% confidence, and all of the correctly classified samples have the minimum confidence scores.

Consider that, to achieve the highest amount of calibration error (100%) for a given set of data requires the following:

$$\boldsymbol{p}_k = 1, \hat{\boldsymbol{p}}_k(\boldsymbol{x}_n) = 1, \boldsymbol{y}_n \neq \hat{\boldsymbol{y}}(\boldsymbol{x}_n) \forall \boldsymbol{x}_n,$$

Based on Eq. 2 , that will satisfy:

$$\mathbb{P}(\boldsymbol{y}_n = \hat{\boldsymbol{y}}(\boldsymbol{x}_n) \mid \hat{\boldsymbol{p}}_k(\boldsymbol{x}_n) = \boldsymbol{p}_k) = 0.$$

In other words, this occurs when a classifier only outputs 100% confidence scores, while its accuracy is 0.

Let a classifier have non-zero accuracy that is unchangeable. We cannot expect to reach the error of 100% since $\boldsymbol{p}_k = 0$ cannot be the case (for the top predicted class) nor can $\mathbb{P}(\boldsymbol{y}_n = \hat{\boldsymbol{y}}(\boldsymbol{x}_n)) = 0$ be true for all $\boldsymbol{y}_n$. However, to achieve the highest calibration error on a set of data points in this scenario, one can first isolate the misclassified data points and if the classifier is made to output confidence scores of 100% on all of them, using calibration attack for example, it would create a total calibration error of 100% on this set of misclassified data points exactly as in the theoretical highest miscalibration scenario discussed previously. Next, with the remaining correctly classified points, which we know the classifier has an accuracy of 100% on, one can create the largest difference between

---

**Algorithm 1** General Calibration Attack

1: **Input:** Classifier $f$, input $\boldsymbol{x}$, original predicted class $\boldsymbol{k}$, $\epsilon$, $I$ attack iterations, attack type $a$
2: **Output:** Adversarial example $\tilde{\boldsymbol{x}}$
3: $\tilde{\boldsymbol{x}} \leftarrow \boldsymbol{x}$
4: **if** a=underconf **then**
5:     Loss function L returns the loss margin between predicted class and second highest class
6: **else if** a=overconf **then**
7:     Loss function L returns softmax probability of predicted class
8: **end if**
9: $l^* \leftarrow L(f(\tilde{x}), k)$
10: **for** $i = 1$ **to** $I$ **do**
11:     Perturb $\tilde{x}$ to find $\delta$ according to attack algorithm bounded by $l_p$ norm e.g. Square Attack
12:     $x_{new} \leftarrow$ Project $\tilde{x} + \delta$ to be within $\epsilon$ of the $l_p$ norm
13:     $l_{new} \leftarrow L(f(x_{new}), k)$
14:     **if** a=underconf and $l_{new} < l^*$ and $argmax(f(x_{new})) = k$ **then**
15:         $\tilde{x} \leftarrow x_{new}, l^* \leftarrow l_{new}$
16:     **else if** a=overconf and $l_{new} > l^*$ **then**
17:         $\tilde{x} \leftarrow x_{new}, l^* \leftarrow l_{new}$
18:     **end if**
19:     **if** ($a$=underconf and $l^* < 0.01$) or ($a$=overconf and $l^* > 0.99$) **then**
20:         break for loop
21:     **end if**
22: **end for**

---

the accuracy and average confidence by making the average confidence on this set as low as possible. Consider Eq. 2, since confidence scores can only range from $1/K$ to 1, the largest possible difference between the average confidence score and the accuracy of 100% is $1 - 1/K$. Again, if $\boldsymbol{p}_k = 1/K$, and every $\hat{\boldsymbol{p}}_k(\boldsymbol{x}_n) = 1/K$, while $y_n = \hat{\boldsymbol{y}}(\boldsymbol{x}_n) \forall \boldsymbol{x}_n$, then this makes the calibration error: $\mathbb{P}(\boldsymbol{y}_n = \hat{\boldsymbol{y}}(x_n) \mid \hat{\boldsymbol{p}}_k(\boldsymbol{x}_n) = \boldsymbol{p}_k) - \boldsymbol{p}_k = 1 - 1/K$. It is not possible to create a higher level of calibration error since if $\boldsymbol{p}_k > 1/K$ on some number of the correctly classified samples, then $\mathbb{P}(y_n = \hat{\boldsymbol{y}}(\boldsymbol{x}_n) \mid \hat{\boldsymbol{p}}_k(\boldsymbol{x}_n) = \boldsymbol{p}_k)$ will still be 1, while $\boldsymbol{p}_k > 1/K$ will lead to less calibration error on that subset of samples. With errors on both mutually exclusive subsets of data maximized, the theoretical highest miscalibration will be created on the full data. Finally, to emulate this scenario, underconfidence attacks are conducted on all data points correctly classified by the model, and overconfidence attacks are conducted on all of the misclassified data yielding the maximum miscalibration attack.

**Random Confidence Attack.** This variation of attacks attempt to completely decouple the model's confidence scores and predictive performance in a randomized fashion by modifying inputs, such that the confidence scores produced by the model on them are randomized. Specifically, random confidence attacks are performed by choosing a random goal confidence score for each input, and then, depending on the original confidence score, running the corresponding underconfidence or overconfidence attacks to produce a new target confidence score on the input. Although this form of attacks are theoretically less effective than the maximum miscalibration attacks, they are less predictable, since unlike the other attacks it does not lead the model to produce one or two terminal confidence scores for each input, and instead can produce more initially reasonable looking confidence scores that are completely meaningless due to being randomized.

## 3.3 DEFENCE AGAINST CALIBRATION ATTACK

As we will show in Section 4, calibration attacks are very effective on existing victim models, hence developing defence strategies is important. In addition to investigating existing defence approaches against regular adversarial attacks, we propose two novel methods specifically against calibration attacks: *Calibration Attack Adversarial Training (CAAT)* and *Compression Scaling (CS)*.

CAAT is a variation of PGD-based adversarial training utilizing our white-box calibration attacks to generate adversarial training samples for each minibatch during training. Hence, both under and overconfident samples with the model's original predicted label preserved are exclusively used to train the model. CS is a post-process scaling technique primarily designed for dealing with maximum miscalibration attacks. It is based on the assumption that since effective classifiers have a high level of accuracy (and confidence), calibration attacks will typically do most of the damage by lowering confidence scores on highly confident correctly classified samples. Therefore, by scaling the low scores to a high confidence range, CS can help mitigate the most severe miscalibration.

For our algorithm, the range of possible confidence scores are first split into $M$ equally sized bins, which we select to be 3 (or 4). Any samples whose confidence scores fall within the bin $m \in \{1, ..., M\}$ in original group $O$ are mapped to a corresponding bin in high confidence bin group $Q$ with a compressed confidence range. In our case, we divide confidence scores into 15 bins and chose the top 3 (or 4) highest confidence bins as the corresponding compressed bins. A linear mapping is applied and the samples' positions in the new bin are kept to be the same as in the original one. Next, the sample's logit vector is scaled by finding a scaling factor by iterating through a large range of possible values so that the new desired confidence score for the sample achieved within the new confidence range. Formally, for a sample $x$ with logit distribution $l = \{l_1, ..., l_k\}$ and original predicted probability $\hat{p}_{og}$, we find temperature $T$ such that it is mapped from a relative bin in bin group $O$ to bin group $Q$. The new predicted probability $\hat{p}_{new}$ is $\arg\max_i \left( \frac{\exp(l_i/T)}{\sum_j \exp(l_j)/T} \right)$ such that $\hat{p}_{new} = \min \text{conf}(Q_m) + \frac{\hat{p}_{og} - \min \text{conf}(O_m)}{\text{range}(O_m)} * \text{range}(Q_m)$, where $\min \text{conf}(Q_m)$ is the minimum confidence level of bin $m$ in $Q$, while $\max \text{conf}(O_m)$ is the maximum confidence level of bin $m$ in $O$, and $\text{range}()$ gives the range of confidence vales in a bin. That is, for bin group $O$ it is $\max \text{conf}(O_m) - \min \text{conf}(O_m)$.

Table 1: Results of four types of $l_\infty$ calibration attacks: underconfidence (*und-atk*), overconfidence (*ovr-atk*), maximum miscalibration (*max-atk*), and random confidence attack (*rnd-atk*). Accuracies of victim models are included.

**ResNet**

| | Avg #q | Med. #q | ECE | KS | Avg. Conf. |
|---|---|---|---|---|---|
| **CIFAR-100** (Accuracy: $0.881_{\pm 0.002}$) | | | | | |
| Pre-atk | - | - | $.052_{\pm.006}$ | $.035_{\pm.006}$ | $.916_{\pm.006}$ |
| Und-atk | $74.3_{\pm3.4}$ | $42.7_{\pm1.5}$ | $.540_{\pm.005}$ | $.479_{\pm.001}$ | $.465_{\pm.005}$ |
| Ovr-atk | $16.0_{\pm0.8}$ | $1.0_{\pm0.0}$ | $.124_{\pm.002}$ | $.124_{\pm.002}$ | $.996_{\pm.000}$ |
| Max-atk | $72.9_{\pm2.8}$ | $41.5_{\pm2.8}$ | $.606_{\pm.002}$ | $.497_{\pm.002}$ | $.502_{\pm.002}$ |
| Rnd-atk | $68.9_{\pm4.6}$ | $42.7_{\pm1.2}$ | $.558_{\pm.011}$ | $.461_{\pm.003}$ | $.514_{\pm.003}$ |
| **Caltech-101** (Accuracy: $0.966_{\pm 0.004}$) | | | | | |
| Pre-atk | - | - | $.035_{\pm.003}$ | $.031_{\pm.004}$ | $.936_{\pm.001}$ |
| Und-atk | $333.8_{\pm13.8}$ | $259.7_{\pm17.4}$ | $.361_{\pm.005}$ | $.362_{\pm.005}$ | $.605_{\pm.006}$ |
| Ovr-atk | $75.7_{\pm9.3}$ | $1.0_{\pm0.0}$ | $.028_{\pm.003}$ | $.028_{\pm.004}$ | $.992_{\pm.000}$ |
| Max-atk | $182.6_{\pm5.6}$ | $286.5_{\pm16.1}$ | $.397_{\pm.008}$ | $.379_{\pm.007}$ | $.618_{\pm.005}$ |
| Rnd-atk | $178.5_{\pm14.9}$ | $289.3_{\pm8.1}$ | $.344_{\pm.014}$ | $.342_{\pm.010}$ | $.638_{\pm.006}$ |
| **GTSRB** (Accuracy: $0.972_{\pm 0.000}$) | | | | | |
| Pre-atk | - | - | $.019_{\pm.006}$ | $.008_{\pm.002}$ | $.980_{\pm.002}$ |
| Und-atk | $197.5_{\pm10.3}$ | $103.0_{\pm7.3}$ | $.396_{\pm.017}$ | $.390_{\pm.013}$ | $.591_{\pm.014}$ |
| Ovr-atk | $12.1_{\pm1.3}$ | $1.0_{\pm0.0}$ | $.029_{\pm.008}$ | $.029_{\pm.008}$ | $.998_{\pm.000}$ |
| Max-atk | $142.1_{\pm6.0}$ | $102.2_{\pm3.6}$ | $.419_{\pm.009}$ | $.402_{\pm.012}$ | $.597_{\pm.011}$ |
| Rnd-atk | $139.4_{\pm1.5}$ | $104.7_{\pm3.5}$ | $.399_{\pm.009}$ | $.386_{\pm.005}$ | $.599_{\pm.007}$ |

**ViT**

| | Avg #q | Med. #q | ECE | KS | Avg. Conf. |
|---|---|---|---|---|---|
| **CIFAR-100** (Accuracy: $0.935_{\pm 0.002}$) | | | | | |
| Pre-atk | - | - | $.064_{\pm.006}$ | $.054_{\pm.005}$ | $.882_{\pm.004}$ |
| Und-atk | $118.5_{\pm2.4}$ | $62.0_{\pm3.1}$ | $.572_{\pm.007}$ | $.553_{\pm.004}$ | $.404_{\pm.003}$ |
| Ovr-atk | $524.7_{\pm88.7}$ | $510.5_{\pm114.3}$ | $.043_{\pm.007}$ | $.043_{\pm.006}$ | $.974_{\pm.001}$ |
| Max-atk | $104.8_{\pm7.5}$ | $62.7_{\pm4.7}$ | $.616_{\pm.003}$ | $.564_{\pm.000}$ | $.431_{\pm.001}$ |
| Rnd-atk | $106.4_{\pm3.0}$ | $70.3_{\pm1.5}$ | $.549_{\pm.002}$ | $.505_{\pm.003}$ | $.471_{\pm.007}$ |
| **Caltech-101** (Accuracy: $0.961_{\pm 0.024}$) | | | | | |
| Pre-atk | - | - | $.137_{\pm.059}$ | $.136_{\pm.060}$ | $.825_{\pm.083}$ |
| Und-atk | $325.5_{\pm16.7}$ | $273.7_{\pm23.7}$ | $.426_{\pm.044}$ | $.426_{\pm.044}$ | $.536_{\pm.068}$ |
| Ovr-atk | $52.1_{\pm40.9}$ | $1.0_{\pm0.0}$ | $.081_{\pm.042}$ | $.079_{\pm.040}$ | $.881_{\pm.067}$ |
| Max-atk | $150.7_{\pm12.1}$ | $269.7_{\pm25.1}$ | $.415_{\pm.036}$ | $.414_{\pm.034}$ | $.551_{\pm.058}$ |
| Rnd-atk | $129.0_{\pm17.3}$ | $315.0_{\pm17.4}$ | $.364_{\pm.016}$ | $.364_{\pm.016}$ | $.598_{\pm.040}$ |
| **GTSRB** (Accuracy: $0.947_{\pm 0.006}$) | | | | | |
| Pre-atk | - | - | $.040_{\pm.005}$ | $.026_{\pm.017}$ | $.922_{\pm.024}$ |
| Und-atk | $169.8_{\pm15.0}$ | $88.3_{\pm6.7}$ | $.459_{\pm.015}$ | $.452_{\pm.019}$ | $.498_{\pm.026}$ |
| Ovr-atk | $94.9_{\pm45.9}$ | $3.7_{\pm4.6}$ | $.029_{\pm.003}$ | $.030_{\pm.004}$ | $.976_{\pm.011}$ |
| Max-atk | $137.1_{\pm4.3}$ | $88.3_{\pm6.7}$ | $.519_{\pm.020}$ | $.480_{\pm.020}$ | $.509_{\pm.024}$ |
| Rnd-atk | $129.5_{\pm7.4}$ | $97.2_{\pm9.9}$ | $.454_{\pm.012}$ | $.432_{\pm.016}$ | $.538_{\pm.019}$ |

## 4 EXPERIMENTAL RESULTS OF CALIBRATION ATTACK

**Metrics.** Two widely used metrics are used to assess calibration error: Expected Calibration Error (ECE) (Pakdaman Naeini et al., 2015) and Kolmogorov-Smirnov Calibration Error (KS error) (Gupta et al., 2021). We evaluate attacks' query efficiency using the average and median number of queries for the attacks to complete (Andriushchenko et al., 2020). Average confidence of predictions is used to judge the degree that the confidence scores are affected. (See Appendix B for details.)

**Datasets.** We use CIFAR-100 (Krizhevsky & Hinton, 2009), Caltech-101 (Fei-Fei et al., 2004), and the German Traffic Sign Recognition Benchmark (GTSRB) (Houben et al., 2013) with respect to safety critical applications.

**Models.** To explore the effect of attacks on different architecture, we use ResNet-50 (He et al., 2016) and the popular non-convolutional attention-based model, the Vision Transformer (ViT) (Dosovitskiy et al., 2021). Details can be seen in Appendices B and C.

**Attack Settings.** Key attack settings, e.g., $l_\infty$, $l_2$ and attack iterations, can be found in Appendix B.

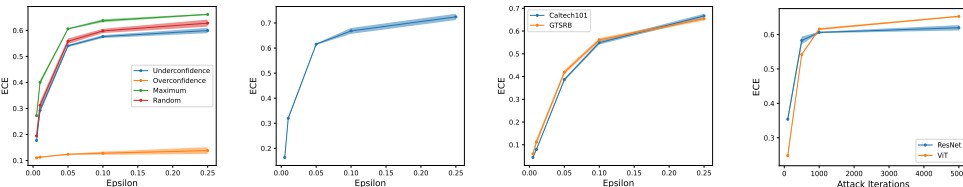

Figure 2: The influence of $\epsilon$ (the left three subfigures) and attack iterations (the right subfigure) in the attacks effectiveness. Subfigure-1 (the left most) presents the comparison between the ECE scores of the different calibration attacks at different $\epsilon$ values using ResNet-50 models trained on CIFAR-100. Subfigure-2: ECE vs. $\epsilon$ using maximum miscalibration attacks on ViT models trained on CIFAR-100. Subfigure-3: ECE vs. $\epsilon$ using maximum miscalibration attacks on the ResNet-50 models trained on Caltech-101 and GTSRB. Subfigure-4: Effect of the numbers of attack iterations on the ability of the attack algorithm. The first three subfigures are created at the $1000^{th}$ iteration.

## 4.1 Overall Performance

The performance of $l_\infty$ attacks on different victim models are shown in Table 1. The results of $l_2$ attacks can be found in Table 6 of the Appendix F.6, and the main results for the white-box attacks are in Table 5 of the Appendix F.1. The experiments show that the maximum miscalibration attack is consistently the most effective in producing the highest levels of miscalibration in both black-box and white-box setups. In most cases its efficiency is competitive, suggesting it has overall the best performance under the same number of attack iterations. The underconfidence and random confidence attack also create substantial calibration error. They perform similarly to one another in terms of overall error, with the underconfidence attacks largely edging in in effectiveness. The results are obtained on three runs of each model with different random seeds.

The overconfidence attack results are of particular interest—the attack successfully raises the average confidence level to around 99% in most cases. Given the very high level of accuracy of each of these models, increasing the confidence level of the predictions does not have as drastic an effect on the calibration error since the gap between the average confidence level and accuracy will still be small when confidence levels are raised to 100%. It can even appear to make the model produce better calibrated scores on the new adversarial data if the model was originally underconfident on the data. However, the effect of the overconfidence attacks is still dangerous since it can make the model output near certain confidence scores irrespective of the input. In practice this can render the use of confidence scores meaningless as they would be the same for every input, so the low calibration error does not necessarily reflect the severity of the attacks. Figure 1 demonstrates the effects of each form of attacks in calibration diagrams, in which we can observe the severity of miscalibration.

The general trends between the $l_2$ and $l_\infty$ attacks are similar, although the latter are more effective, leading to more severe miscalibration whilst being more query efficient and faster by as much as two or three times. There are only a few cases where $l_2$ attacks yield higher miscalibration or more efficient attacks in terms of the number of queries. We focus our remaining analyses on the more effective $l_\infty$ norm. Regarding different architectures, the ViT models tend to be slightly more miscalibrated after the attacks, compared to the ResNet models on ECE and KS, but the ViT models also are notably more miscalibrated before the attacks.

Even with the relatively low amount of queries under the black-box setting, all of the types of attacks are generating severe miscalibration, increasing ECE and KS error often by over 10 times the original amount. Calibration attacks are very effective in achieving their goals even with the prediction accuracy unchanged, raising serious concerns for safety critical applications.

## 4.2 Detection Difficulty Analysis

Calibration attacks are difficult to detect given they do not affect the decisions of the classifier. To show this, we run experiments using ResNet-50 on our three tested datasets using the popular adversarial attack detection methods: Local Intrinsic Dimensionality (LID) (Ma et al., 2018), Mahalanobis Distance (MD) (Lee et al., 2018), and SpectralDefense (Harder et al., 2021). The details behind the settings for each detection method can be seen in Appendx E. We show our main results

Table 2: Adversarial attack detection results comparing original version of PGD and SA attacks with their calibration attack counterparts.

| | PGD | | PGD-Under | | PGD-Over | | PGD-Max | | SA | | SA-Under | | SA-Over | | SA-Max | |
|---|---|---|---|---|---|---|---|---|---|---|---|---|---|---|---|---|
| | AUC | Acc | AUC | Acc | AUC | Acc | AUC | Acc | AUC | Acc | AUC | Acc | AUC | Acc | AUC | Acc |
| **CIFAR-100** | | | | | | | | | | | | | | | | |
| LID | 93.7 | 87.7 | 64.5 | 67.1 | 88.7 | 84.3 | 63.3 | 66.0 | 90.1 | 82.9 | 54.2 | 54.3 | 63.9 | 61.6 | 54.5 | 54.0 |
| MD | 99.3 | 98.5 | 83.1 | 74.4 | 96.7 | 93.8 | 81.9 | 73.3 | 99.8 | 98.9 | 90.2 | 80.5 | 78.1 | 74.8 | 89.4 | 79.9 |
| Spect. | 100.0 | 100.0 | 74.9 | 67.0 | 94.2 | 90.0 | 64.8 | 62.5 | 100.0 | 98.0 | 70.5 | 65.5 | 52.3 | 50.5 | 71.6 | 65.5 |
| **Caltech-101** | | | | | | | | | | | | | | | | |
| LID | 84.8 | 78.1 | 53.6 | 54.8 | 89.1 | 81.9 | 53.9 | 55.1 | 70.2 | 64.4 | 62.5 | 61.1 | 65.3 | 62.3 | 58.6 | 59.5 |
| MD | 91.6 | 84.6 | 60.1 | 57.9 | 90.6 | 84.7 | 62.6 | 60.3 | 88.9 | 81.3 | 81.9 | 74.7 | 73.9 | 70.1 | 81.8 | 74.5 |
| Spect.l | 93.4 | 88.5 | 67.8 | 66.5 | 93.7 | 90.5 | 64.2 | 62.0 | 98.0 | 94.5 | 59.1 | 53.0 | 51.8 | 50.0 | 56.9 | 53.0 |
| **GTSRB** | | | | | | | | | | | | | | | | |
| LID | 95.7 | 89.1 | 88.8 | 86.3 | 94.6 | 87.4 | 87.0 | 85.3 | 86.3 | 77.1 | 71.4 | 68.1 | 72.5 | 69.8 | 72.4 | 66.7 |
| MD | 100.0 | 99.8 | 94.6 | 93.6 | 97.9 | 97.7 | 92.9 | 92.6 | 95.5 | 89.6 | 83.7 | 77.8 | 74.4 | 74.9 | 85.6 | 79.0 |
| Spect. | 99.4 | 98.5 | 94.8 | 93.0 | 99.0 | 99.9 | 99.0 | 99.9 | 99.1 | 98.0 | 83.0 | 79.0 | 50.4 | 50.5 | 83.1 | 79.0 |

of the effectiveness of the detectors in terms of Area Under the Curve (AUC) and detection accuracy when running the different types of calibration attacks compared to their original counterparts in Table 2, under both the white-box and black-box setups. We observe how in most cases, with one exception, there is a notable decrease in detection performance, with the effects being most pronounced in SA, where the decrease of more than 20% in accuracy can often be seen. Existing detection methods appear less reliable for calibration attacks.

## 4.3 IMPACT OF KEY ASPECTS OF ATTACK

We examine the impact of key aspects of calibration attacks, including adversarial noise, attack iterations, and attack types. More details can be found in Appendix F, including additional analyses. All the analyses are conducted over the more effective $l_\infty$ attacks in the black-box setup using our regular attack settings unless stated otherwise.

**Epsilon.** We study how successful the different attacks are at different $\epsilon$ levels to ascertain how low the value can be in order to have notable harm. The results are shown in Figure 2. Our first set of experiments seen in the leftmost figure is based on CIFAR-100 using ResNet-50, where we track the ECE at five different $\epsilon$ values after being attacked using all the different attacks. The attacks are successful even with low $\epsilon$ values. The rise in ECE is sharp when $\epsilon$ increases, but it plateaus at higher values. The middle figure shows similar results when using ViT with the maximum miscalibration attack, revealing that the phenomena is not architecture dependent. The rightmost figure, where a maximum miscalibration attack is used against ResNet over both Caltech-101 and GTSRB, shows similar patterns across different datasets.

Table 3: Comparison between the efficiency of the underconfidence (shortened as *und.*) and overconfidence (shortened as *ovr.*) attacks.

| **ResNet** | | | | |
|---|---|---|---|---|
| | # Samples | Avg. #q | Med. #q | Avg. Conf. |
| **CIFAR-100** | | | | |
| 90% -10% *und.* | $186.0_{\pm4.4}$ | $8.5_{\pm0.5}$ | $5.7_{\pm0.6}$ | $77.9_{\pm1.4}$ |
| 90% +10% *ovr.* | $186.0_{\pm4.4}$ | $36.5_{\pm1.5}$ | $31.0_{\pm1.7}$ | $99.0_{\pm0.0}$ |
| 80% -10% *und.* | $94.0_{\pm4.2}$ | $6.5_{\pm0.5}$ | $4.3_{\pm0.7}$ | $67.9_{\pm0.4}$ |
| 80% +10% *ovr.* | $94.0_{\pm4.2}$ | $9.8_{\pm1.7}$ | $7.0_{\pm1.4}$ | $91.3_{\pm0.1}$ |
| **Caltech-101** | | | | |
| 90% -10% *und.* | $160.7_{\pm7.8}$ | $39.4_{\pm11.8}$ | $31.7_{\pm8.5}$ | $80.2_{\pm0.0}$ |
| 90% +10% *ovr.* | $160.7_{\pm7.8}$ | $225.6_{\pm55.5}$ | $212.8_{\pm63.3}$ | $98.8_{\pm0.2}$ |
| 80% -10% *und.* | $53.3_{\pm24.0}$ | $17.5_{\pm12.1}$ | $16.0_{\pm19.1}$ | $63.1_{\pm2.1}$ |
| 80% +10% *ovr.* | $53.3_{\pm24.0}$ | $30.0_{\pm10.3}$ | $21.7_{\pm11.3}$ | $89.5_{\pm0.4}$ |
| **GTSRB** | | | | |
| 90% -10% *und.* | $48.0_{\pm4.2}$ | $11.7_{\pm0.1}$ | $8.2_{\pm1.4}$ | $77.5_{\pm1.2}$ |
| 90% +10% *ovr.* | $48.0_{\pm4.2}$ | $98.5_{\pm4.0}$ | $62.8_{\pm5.7}$ | $99.1_{\pm0.0}$ |
| 80% -10% *und.* | $30.0_{\pm0.7}$ | $9.2_{\pm0.9}$ | $6.0_{\pm1.1}$ | $70.5_{\pm0.7}$ |
| 80% +10% *ovr.* | $30.0_{\pm0.7}$ | $18.6_{\pm3.4}$ | $12.8_{\pm0.4}$ | $91.0_{\pm0.1}$ |
| | | | | |
| **ViT** | | | | |
| **CIFAR-100** | | | | |
| 90% -10% *und.* | $392.7_{\pm33.5}$ | $20.0_{\pm1.6}$ | $9.3_{\pm1.2}$ | $76.1_{\pm0.3}$ |
| 90% +10% *ovr.* | $392.7_{\pm33.5}$ | $883.7_{\pm120.2}$ | $883.7_{\pm120.2}$ | $97.1_{\pm0.1}$ |
| 80% -10% *und.* | $138.7_{\pm5.1}$ | $8.8_{\pm1.7}$ | $5.5_{\pm0.5}$ | $66.1_{\pm0.3}$ |
| 80% +10% *ovr.* | $138.7_{\pm5.1}$ | $55.8_{\pm1.4}$ | $20.7_{\pm4.7}$ | $90.0_{\pm0.1}$ |
| **Caltech-101** | | | | |
| 90% -10% *und.* | $472.0_{\pm161.0}$ | $264.7_{\pm56.6}$ | $198.0_{\pm51.0}$ | $81.1_{\pm0.5}$ |
| 90% +10% *ovr.* | $472.0_{\pm161.0}$ | $700.7_{\pm518.5}$ | $700.7_{\pm518.5}$ | $93.2_{\pm0.2}$ |
| 80% -10% *und.* | $222.0_{\pm17.3}$ | $149.5_{\pm19.3}$ | $66.5_{\pm9.8}$ | $70.0_{\pm0.6}$ |
| 80% +10% *ovr.* | $222.0_{\pm17.3}$ | $323.9_{\pm123.5}$ | $269.3_{\pm210.7}$ | $86.0_{\pm0.2}$ |
| **GTSRB** | | | | |
| 90% -10% *und.* | $184.0_{\pm27.5}$ | $31.0_{\pm6.5}$ | $12.7_{\pm1.9}$ | $76.1_{\pm0.9}$ |
| 90% +10% *ovr.* | $184.0_{\pm27.5}$ | $235.3_{\pm69.5}$ | $180.0_{\pm92.0}$ | $96.5_{\pm0.2}$ |
| 80% -10% *und.* | $79.0_{\pm28.6}$ | $16.1_{\pm6.7}$ | $7.7_{\pm2.3}$ | $67.3_{\pm0.2}$ |
| 80% +10% *ovr.* | $79.0_{\pm28.6}$ | $140.1_{\pm63.2}$ | $41.7_{\pm43.7}$ | $89.6_{\pm0.8}$ |

**Iterations.** Creating attacks that are effective without requiring a large number of queries is important. Figure 2 shows the results of varying the number of iterations using the maximum miscalibration attack on the two tested models on CIFAR-100. We find that even at 100 iterations the ResNet model becomes heavily miscalibrated. In Appendix F.2, we also provide detailed comparison on the effectiveness of the maximum miscalibration attack to the base SA algorithm. We find that our attacks consistently produces higher amounts of miscalibration compared to the original SA across different number of iterations, showing how effective calibration attacks are at specifically targeting creating miscalibration, and how optimizing targeting accuracy and calibration are separate objectives and worth considering independently.

**Underconfidence vs. Overconfidence.** To understand which form of attack is most query efficient when the amount of change in confidence is the same, for each attack type we identify all of images in the test set that are around a given confidence level. We use the corresponding attack to made the model produce either an increase of 10% in confidence, or a decrease of 10%. We choose two base confidence levels of 80% and 90% and find all the data points within 1% of each. When an attack causes a change at or past the set threshold for the given goal probability, the attack stops and the number of queries is recorded. The results can be seen in Table 3. The consistent pattern we observe for both base confidence levels is that it takes notably fewer queries to create underconfidence than overconfidence, and the former attack is more effective at affecting the average confidence.

**Qualitative Analysis.** We perform qualitative analyses on the effect of calibration attacks. Appendix F.3 provides t-SNE visualization for calibration-attacked data samples. In Figure 3 below, we demonstrate the difference in the coarse localization maps using the GradCAM technique (Selvaraju et al., 2017) for images before and after calibration attack (see Appendix F.5 for details). In the first case, we confirm that all of the attacks are leading to the expected behaviour in the representation space, with the overconfidence attacks in particular being successful in pushing affected samples far from the decision boundary. With the Grad-CAM visualizations, we find that there are minimal changes to the maps after the attacks are conducted, making it difficult to identify using gradient visualization methods.

Table 4: Effectiveness of calibration methods and adversarial defences, evaluated with pre- and post-ECE and KS scores.

|  | Avg#q | Med#q | Acc | PrECE | PsECE | PrKS | PsKS |
|---|---|---|---|---|---|---|---|
| **WideResNet (CIFAR-100)** | | | | | | | |
| Gowal, '20 | 63.5 | 86.5 | .690 | .137 | .248 | .137 | .200 |
| Rebuffi, '21 | 36.4 | 51.0 | .622 | .190 | .209 | .189 | .198 |
| Pang, '22 | 50.5 | 64.5 | .638 | .185 | .214 | .187 | .195 |
| **ResNet-50** | | | | | | | |
| **CIFAR-100** | | | | | | | |
| TS | 74.1 | 40.0 | .880 | .034 | .643 | .007 | .530 |
| Splines | 6.6 | 83.0 | .876 | **.020** | .681 | .019 | .573 |
| DCA | 68.0 | 39.0 | .866 | .049 | .604 | .039 | .492 |
| SAM | 83.8 | 44.5 | .882 | .033 | .609 | .014 | .506 |
| AAA | 7.7 | 31.5 | .880 | .038 | .225 | .011 | .123 |
| AT | 65.9 | 60.0 | .790 | .035 | .431 | .022 | .279 |
| CAAT | 66.9 | 44.0 | .842 | .048 | .504 | .036 | .440 |
| CS | 64.4 | 39.0 | .880 | .051 | **.218** | .041 | .145 |
| **Caltech-101** | | | | | | | |
| TS | 194.7 | 276.0 | .970 | **.014** | .347 | .005 | .322 |
| Splines | 4.5 | 150.0 | .970 | .019 | .104 | .010 | .095 |
| DCA | 189.6 | 269.0 | .962 | .038 | .418 | .027 | .392 |
| SAM | 191.1 | 276.0 | .970 | .051 | .429 | .049 | .414 |
| AAA | 1.1 | 20.0 | .964 | .061 | .100 | .058 | .085 |
| AT | 23.7 | 194.0 | .918 | .038 | .079 | .018 | .068 |
| CAAT | 127.5 | 206.0 | .972 | .017 | .264 | .012 | .266 |
| CS | 179.4 | 254.0 | .970 | .026 | **.065** | .017 | .067 |
| **GTSRB** | | | | | | | |
| TS | 160.6 | 111.0 | .972 | .019 | .396 | .018 | .377 |
| Splines | 0.7 | 22.0 | .972 | .018 | .129 | .007 | .123 |
| DCA | 130.2 | 97.0 | .976 | .017 | .389 | .011 | .372 |
| SAM | 117.4 | 87.0 | .978 | **.012** | .384 | .003 | .371 |
| AAA | 3.1 | 51.0 | .972 | .023 | **.071** | .014 | .059 |
| AT | 37.7 | 117.5 | .962 | .017 | .160 | .007 | .135 |
| CAAT | 121.7 | 115.0 | .968 | .020 | .324 | .017 | .317 |
| CS | 151.2 | 111.0 | .972 | .019 | .097 | .020 | .095 |
| **ViT** | | | | | | | |
| **CIFAR-100** | | | | | | | |
| TS | 117.7 | 73.0 | .938 | **.014** | .568 | .010 | .515 |
| Splines | 9.9 | 130.5 | .938 | .023 | .405 | .016 | .358 |
| DCA | 125.6 | 69.0 | .944 | .024 | .565 | .011 | .519 |
| SAM | 123.0 | 66.0 | .942 | .072 | .607 | .064 | .561 |
| AAA | 0.8 | 42.0 | .938 | .106 | .200 | .092 | .161 |
| AT | 86.8 | 77.0 | .886 | .066 | .519 | .063 | .439 |
| CAAT | 102.4 | 56.0 | .922 | .026 | .537 | .010 | .506 |
| CS | 97.0 | 59.0 | .938 | .044 | **.137** | .033 | .142 |
| **Caltech-101** | | | | | | | |
| TS | 154.8 | 280.0 | .972 | **.030** | .313 | .023 | .264 |
| Splines | 0.5 | 45.0 | .972 | .035 | .071 | .017 | .049 |
| DCA | 140.9 | 254.5 | .976 | .039 | .345 | .025 | .345 |
| SAM | 145.9 | 278.0 | .962 | .170 | .459 | .170 | .459 |
| AAA | 0.3 | 20.5 | .972 | .189 | .196 | .189 | .198 |
| AT | 48.2 | 188.0 | .946 | .132 | .229 | .132 | .231 |
| CAAT | 136.2 | 341.0 | .986 | .048 | .316 | .049 | .318 |
| CS | 143.3 | 277.0 | .934 | .205 | **.068** | .018 | .068 |
| **GTSRB** | | | | | | | |
| TS | 132.9 | 80.0 | .950 | **.038** | .463 | .033 | .410 |
| Splines | 1.7 | 16.0 | .950 | .040 | .115 | .041 | .067 |
| DCA | 132.8 | 92.0 | .950 | .052 | .506 | .037 | .476 |
| SAM | 133.9 | 103.0 | .944 | .070 | .505 | .069 | .473 |
| AAA | 0.1 | 4.0 | .950 | .053 | .128 | .049 | .110 |
| AT | 66.9 | 124.0 | .930 | .132 | .320 | .130 | .317 |
| CAAT | 118.9 | 85.0 | .932 | .066 | .446 | .055 | .431 |
| CS | 130.5 | 81.5 | .950 | .027 | **.092** | .035 | .089 |

# 5 RESULTS OF DEFENDING AGAINST CALIBRATION ATTACKS

In this section, we investigate 11 types of defence based on common or effective adversarial defences and calibration techniques. We compare the performance to the two methods we propose. A complete description with training details for each method is in Appendix C.2.

**Baseline Defence Methods.** For the calibration-focused models, we use two popular and highly effective post-calibration methods, Temperature Scaling (TS) (Guo et al., 2017) and Calibration using Splines (Gupta et al., 2021). For training-based regularization methods we use two effective models, DCA (Liang et al., 2020) and SAM (Foret et al., 2021). Regarding adversarial defence methods, we test the top-3 state-of-the-art defence models under the $l_\infty$ attack for CIFAR-100, using WideResNet on the RobustBench leaderboard (Croce et al., 2021). We include one of the most common defences in the form of a PGD-based adversarial training (AT) (Madry et al., 2018) baseline, which is tuned to be well calibrated on clean data, and a recent post-process defense called Adversarial Attack Against Attackers (AAA) (Chen et al., 2022), due to its effectiveness against SA.

**Results and Analyses.** The results for the comparison between the different methods can be seen in Table 4. We show the results of the best performing models trained in each case, and we measure query efficiency, accuracy, along with the ECE and KS error, before and after the attack. The attack model is the $l_\infty$ maximum miscalibration attack using the same settings described in Appendix B.

The RobustBench models compromise substantially on accuracy, and have a high level of miscalibration on clean data. They do largely avoid getting extremely miscalibrated as a result of the attacks compared to the defenceless models, except the top model on the leaderboard. Nevertheless, the high inherent miscalibration of these classifiers means they are unfavourable in situations where the model must be well calibrated.

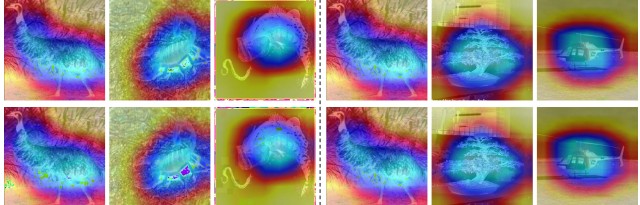

Figure 3: GradCAM visualizations on samples showing the image regions most responsible for the decisions of ResNet-50 before (top row) and after being attacked by the underconfidence and overconfidence attacks (bottom row). The left set of images shows the underconfidence attack visualizations and the right displays the overconfidence attack visualizations. Examples are picked to have large confidence differences after being attacked.

In terms of the calibration methods, TS tends to be among the best methods at reducing calibration error prior to the attacks, but after the attacks it offers very little, if any, robustness compared to the vanilla models. The Splines method is similar in its pre-attack calibration benefits, but differs greatly in its performance post-attack. In some cases, like CIFAR-100 ResNet, it is easily the worst performing defence method. In other cases, particularly for Caltech-101 and GTSRB ViT, it is able to keep ECE at relatively reasonable values post-attack. This discrepancy shows that finding an ideal recalibration function has the potential to be a strong defence. The training-based DCA and SAM methods tend to bring few benefits after being attacked, even when they improve the calibration on clean data, the post-attack ECE and KS errors are not substantially different compared to the vanilla models.

The performance of the regular adversarial defence techniques is mixed. In terms of robustness, AAA is in most cases the technique with among the lowest post-attack ECE. Even in the best cases like Caltech-101 ResNet, the ECE tends to be at least double compared pre-attack, and in most cases we still observed multiple-fold increases. This technique is also among the poorest calibrated on clean data. Regarding AT, our approach does not compromise on accuracy and miscalibration on clean data. Likewise, it does bring some notable robustness, especially compared to the calibration methods, but it is not among the strongest.

Lastly, CAAT tends to perform more poorly in resisting the attacks than AT, likely because the training samples do not deceive the model so there is less inductive bias towards general adversarial robustness. In contrast, the performance of CS is quite strong, and it is the strongest method at maintaining low calibration error post-attack overall, even compared to the adversarial-based methods. Moreover, the technique tends to have better calibration error on clean data compared to AAA. It shows how it is key that high confidence values are retained to have decent calibration after the attacks. Altogether, despite some promising results with the defences, as a whole there are still limitations particularly with the strongest adversarial defences. The compromise of poor ECE on clean data for better calibration robustness against the attacks that we observe, as well as the general inconsistent performance means that further refinement on defences is warranted.

## 6 CONCLUSION AND OUTLOOK

We introduced and studied a new form of adversarial attack that targets the calibration of deep models. Specifically, we developed four forms of calibration attacks and showed the danger of them on typical victim models, and also on models using current calibration and defence methods. We empirically showed that calibration attacks can present a major problem for safety critical systems, as they can be more difficult to detect than standard attacks, and if the calibration of the model on the data is not being closely monitored it is easy to be unaware that the model is not being attacked since the accuracy is unchanged. Also, we devised two novel defences against such calibration attacks, and empirically showed their effectiveness.

Our study here indicates that creating models that are robust against such calibration attacks through different defence methods is of tantamount importance for future work, especially with the mixed performance of various techniques we have observed, and because creating theoretically stronger and more efficient attacks is possible, particularly in terms of the overconfidence attack.

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

## A    A MORE DETAILED SUMMARY OF RELATED WORK

In the field of calibration, a great deal of current research is devoted to the creation of new calibration methods that can be applied to create better calibrated models while possessing as minimum overhead in applying them as possible. Methods are generally divided into two types. Post-calibration methods can be applied directly to the predictions of fully trained models at test time, and methods of this class include temperature scaling (Guo et al., 2017). More traditional methods of this type include Platt scaling (Platt, 1999), isotonic regression (Zadrozny & Elkan, 2002), and histogram binning (Zadrozny & Elkan, 2001). All of these three methods are originally formulated for binary classification settings, and work by creating a function that maps predicted probabilities based on their values more in tune with the model's level of performance. Although they are easy to apply, they often come with the limitation of needing a large degree of validation data to tune, especially with isotonic regression, and performance can struggle when applied to more out of distribution data.

The second class of methods are training-based methods, which typically add a bias during training to ensure that a model learns to become better calibrated. Often times these methods help by acting as a form of regularization that can punish high levels of overconfidence late into training. In computer vision, Mixup (Zhang et al., 2018) is a commonly used method of this type that serves as an effective regularizer by convexly combining random pairs of images and their labels and helps calibration primarily due to the use of soft, interpolated labels (Thulasidasan et al., 2019). Other methods work by adding a penalty to the loss function, like in the case of MMCE, an RKHS kernel-based measure of calibration that is added as a penalty on top of the regular loss during training so that both are optimized jointly (Kumar et al., 2018). Similarly, Tomani & Buettner (2021) create a new loss term called adversarial calibration loss that directly minimizes calibration error using adversarial samples. Given the effectiveness of many of these methods in regular testing scenarios, we desire to illustrate how well a diverse range of these methods can cope against attacks targeting model calibration and whether they possess limitations that require them to be overhauled to deal with an attack scenario.

With respect to adversarial attacks, attacks in this field are wide ranging. Well known white-box attacks include the basic Fast Gradient Sign Method (FGSM) (Goodfellow et al., 2015). This method works by finding adjustments to the input data that maximizes the loss function, and uses the back-progogated gradients to produce the adversarial examples. Projected gradient descent (PGD) (Madry et al., 2018) is popular iterative-based method that similarly uses gradient information, and has been shown to be a universal first-order adversary, and thus is the strongest form of attacks making using of gradient and loss information. In the black-box space of attacks, ZOO (Chen et al., 2017) is an example of a popular score-based attack that uses zeroth order stochastic coordinate descent to attack the model, and avoids training a substitute model. The authors make use of attack-space dimension reduction, hierarchical attacks and importance sampling to make the attack more query efficient, which is required as black-box attacks generally need a lot of queries to run compared to white-box methods.

A broad range of defences against adversarial attacks have been developed, but among the most popular and effective is adversarial training (Goodfellow et al., 2015), where during training the loss is minimized over one of or both clean and generated adversarial examples. Adversarial training however greatly increases training time due to the need to fabricate adversarial examples for every batch. Gradient masking (Carlini & Wagner, 2017) is a simple defence based on obfuscating gradients so that attacks cannot make use of gradient information to create adversarial examples, although it can easily be circumvented in many cases for white-box models (Athalye et al., 2018), and black box attacks do not need gradient information in the first place. It is by and large difficult for adversarial defences to keep pace with the broad range of attacks and to be provably robust against a large number of them. Although the main topic of this work is calibration, we do focus on modelling adversarial defences and their effectiveness against our new form of attack.

## B    DETAILS OF EXPERIMENTAL SETUP

**Metrics.**    To assess the degree of calibration error caused by each attack, we use two metrics, the popular binning-based Expect Calibration Error (ECE) (Pakdaman Naeini et al., 2015), and

Kolmogorov-Smirnov Calibration Error (KS error) (Gupta et al., 2021), which are formulated in detail in Section D.

**Datasets.** The datasets we use in our study are CIFAR-100, Caltech-101, and the German Traffic Sign Recognition Benchmark (GTSRB). CIFAR-100 and Caltech-101 are both popular image recognition benchmark datasets, containing various objects divided into 100 classes and 101 classes respectively. Given the importance of calibration in safety critical applications, we include a common use case of autonomous driving with the GTSRB dataset, which consists of images of traffic signs divided 43 classes. CIFAR-100 has 50,000 images for training, and 10,000 for testing. Caltech-101 totals around 9000 images. GTSRB is split into 39,209 training images and 12,630 test images

**Models.** ResNet-50 (He et al., 2016) is primary model we train and test on due to it being a standard model for image classification. Non-convolutional attention-based networks have recently attained great results on image classification tasks, so we also experiment with the popular Vision Transformer (ViT) architecture (Dosovitskiy et al., 2021). Both of these models are the versions with weights pretrained on ImageNet (Deng et al., 2009). We use the VIT_B_16 variant of ViT, and the pretraining dataset used for each model is ImageNet_1K for ResNet and ImageNet_21K for ViT, and are fine-tuned on the target datasets. Pretrained models are advantageous to study given they can increase performance over training from randomly initialized weights and is a more practical use-case. The specific details behind our training procedures and our various model hyperparameters can be seen in Section C.

**Attack Settings.** Regarding the SA version of the attacks, for the $l_\infty$ and $l_2$ norm attacks we use the default SA settings for $\epsilon$ and $p$, which are $\epsilon = 0.05$ and $p = 0.05$ for $l_\infty$ and $\epsilon = 5.0$ and $p = 0.1$ for $l_2$. For our primary results we run the attacks on a representative 500 test samples from the test set of each dataset. Each attack is ran for 1000 iterations, far less than the default 10,000 in Andriushchenko et al. (2020), but since there is no need to change the label, less iterations are required, bolstering the use-case and threat for this form of attacking.

The settings for the PGD version of the attacks differ due to the accommodations that need to be made to prevent the PGD algorithm from changing the label while still being able to have a large effect on the confidence. In terms of general settings, we again use $\epsilon = 0.05$ as the adversarial noise value for an $l_\infty$ norm. We use an $\alpha$ attack step size value of $5/255$. For our white-box results we use 10 iterations of the attack. In addition to these settings, some were made to the attack algorithm as simply preventing PGD from changing the class label while trying to calculate the adversarial noise often leads to poor performance in practice as many updates are prevented. Instead, a dropout factor is added to the $(h * w * c)$ adversarial noise matrix after each attack iteration that only applies a select portion of the updates, lessening the effect of updates that are too strong and have a high chance of flipping the label. The value for the dropout is dependent on whether it is the overconfidence or underconfidence attack. The most effective values in our experiments were found to be a dropout value of 0.95 for the underconfidence attack, and 0.2 for the overconfidence attack.

## C    SPECIFIC TRAINING DETAILS

### C.1    GENERAL SETTINGS

As mentioned previously, for our general attack implementation we use Square Attack, which works by using a randomized search scheme to find localized square-shaped perturbation at random positions which are sampled in such a way as to be situated approximately at the boundary of the feasible set. We still use the original sampling distributions, however we remove the initialization (initial perturbation) for each attack since it is prone to changing the predicted labels. Naturally, we use the untargeted versions of the attacks, whereby the perturbations lead to increases in the probabilites of random non-predicted classes for the underconfidence attack, since we only care about the probability of the top predicted class.

The details of the training procedure for each of the models and datasets is as follows: For CIFAR-100 and GTSRB, we use the predefined training and test sets for both but use 10% of the training data for validation purposes. For Caltech-101, which comes without predetermined splits, we use an 80:10:10 train/validation/test split. For all of the datasets, we resize all images to be 224 by 224. We also normalize all of the data based on the ImageNet channel means and standard deviations. We

apply basic data augmentation during training in the form of random cropping and random horizontal flips to improve model generalizability. The hyperparameters we used for training the ResNet-50 models include: a batch size of 128, with a CosineAnnealingLR scheduler, 0.9 momentum, 5e-4 weight decay, and a stochastic gradient descent (SGD) optimizer. For ViT, the settings are the same, except we also use gradient clipping with the max norm set to 1.0. We conduct basic grid search hyperparameter tuning over a few values for the learning rate (0.1,0.01,0.005,0.001) and training duration (in terms of epochs). Generally, we found that a learning rate of 0.01 worked best for both types of models. The training times vary for each dataset and model. For the ResNet-50 models we trained for 15 epochs on CIFAR-100, 10 epochs on Caltech-101, and 7 epochs on GTSRB. Likewise for ViT, we trained for 10 epochs on CIFAR-100, 15 epochs on Caltech-101, and 5 epochs on GTSRB. The results reported in Sections 3 and 5 are shown for models on the epoch at which they attained the best accuracy on the validation set. We use the TorchVision (Paszke et al., 2019) implementation of ResNet-50, and the Huggingface Transformers (Wolf et al., 2019) implementation of ViT. All of the training occurred on 24 GB Nvidia RTX-3090 and RTX Titan GPUs. Finally, we use 15 bins to calculate the ECE.

## C.2 Defense Training Settings

In this section, we describe each of the defences we used in Section 5, and the settings we use to train them (if applicable).

**Temperature Scaling (TS) (Guo et al., 2017).** TS is a post-process recalibration technique applied to the predictions of an already trained model that reduces the amount of high confidence predictions without affecting accuracy. TS works by re-scaling the logits after the final layer of the neural network to have a higher entropy by dividing them by a temperature parameter $T$, that is tuned by minimizing negative log likelihood (NLL) loss on the validation set. Temperature scaling only works well when the training and test distributions are similar (Kumar et al., 2019), but by reducing overconfidence it may have an advantage against overconfidence attacks.

**Calibration of Neural Networks using Splines (Spline) (Gupta et al., 2021).** Spine is another post-process recalibration technique that uses a recalibration function to map existing neural network confidence scores to better calibrated versions by fitting a spline function approximates the empirical cumulative distribution. It is lightweight, and often performs better than TS.

**Difference between confidence and accuracy (DCA) (Liang et al., 2020).** DCA is a training-based calibration method that adds an auxiliary loss term to the cross-entropy loss during training that penalizes any difference between the mean confidence and accuracy within a single batch, inducing a model to not produce confidence scores that are miscalibrated. We set the weight of DCA to 10 based on the recommendation by Liang et al. (2020). Training settings are kept the same as described in the general settings.

**Sharpness Aware Minimization (SAM) (Foret et al., 2021).** SAM is a technique that improves model generalizability by simultaneously minimizing loss value and loss sharpness. It finds parameters that lie in neighbourhoods having uniformly low loss by computing the regularized "sharpness-aware" gradient. The motivation behind using this technique as a defence is that models with parameters that lie in uniformly low loss areas may be harder to create adversarial examples, and may be more regularized. We use a neighbourhood size $\rho = 0.05$. We kept the training settings the same as we described in the general settings.

**RobustBench (Croce et al., 2021).** To understand how state-of-the-art adversarial defences work against our attack, we take the top 3 performing (in terms of adversarial robustness) WideResNet (Zagoruyko & Komodakis, 2016) defences on the popular RobustBench defence model benchmark for CIFAR-100 under the $l_\infty$ $\epsilon = 8/255$ attack model. We only choose the WideResNet models given their closer similarity to the primary model we study in this work, ResNet-50. The defences we choose are those of Gowal et al. (2020) (ranked first), Rebuffi et al. (2021) (ranked third) and Pang et al. (2022) (ranked fifth). These defences use a combination of adversarial training and ensembling to produce models that are robust against a wide range of conventional adversarial attacks. In addition, they use different techniques, like combining larger models, using Swish/SiLU activations and model weight averaging, and data augmentation to significantly improve robust accuracy.

**Adversarial Training (AT).** AT is among the most common and effective defences against a wide range of adversarial attacks where models are trained on adversarially attacked images. We implement our version similar to Madry et al. (2018) and Xie et al. (2019), where we run PGD-based adversarial training, given how this form of defence has been shown to be effective across a wide range of attacks due to PGD being close to a universal first-order $l_\infty$ attack. We train exclusively on images attacked with an n-step $l_\infty$ PGD attack each batch, with the number of steps chosen depending on the model and dataset. Since we already test RobustBench models that often make use of AT with a large amount of steps, we specifically tune our AT models to have less steps to compromise less on accuracy and miscalibration. We wish to see whether more lightly-tuned AT can still provide major benefits given calibration attacks are not as severe. For the PGD attack, we attack each image in a batch using an $\epsilon$ norm of 0.1. We use an attack stepsize relative to $\epsilon$ of 0.01 / 0.3, with random starts. The number of attack iterations ran for each batch was carefully chosen to balance performance and adversarial robustness. We used 15 iterations on all of the ResNet models, while for ViT we generally required much fewer, with three for the CIFAR-100 models, and five for the remaining two datasets. In terms of remaining training details, we keep them largely the same as described in the general settings, although the training durations were sometimes varied by a few epochs to optimize accuracy. We use the Foolbox implementation of the PGD attack (Rauber et al., 2020; 2017).

**Adversarial Attack Against Attacks (AAA) (Chen et al., 2022).** A recent adversarial defence specifically tuned towards black box score based methods like Square Attack, this is a post processing method that works on an already trained neural network's logits that uses a function that misleads the attack methods towards incorrect attack directions by slightly modifying the output logits. The method is shown to be very effective against score-based query methods at a low computational cost, and is purported to maintain good calibration, which makes it of particular interest in this case as a defence against calibration attacks.

**Calibration Attack Adversarial Training (CAAT).** Our novel form of adversarial training that uses calibration attacks to generate adversarial samples rather than the regular attack algorithm. Although the general methodology is still the same as PGD-based adversarial training, the primary difference is that for each minibatch, both the underconfidence PGD calibration attack and its overconfidence version are applied to the images and the loss between the two sets of images is added. As this uses calibration attack, the labels of these images are unaffected. The settings we use for the attacks are the same as those described in B for the white-box version. Regarding the settings for each model and dataset, they are largely similar to those of regular AT, although the number of attack iterations is kept consistent at 10, even for ViT. The number of training epochs are the same as those we use for regular fine-tuning.

**Compression Scaling (CS).** This is a novel post-process defence that does not require training and is specifically designed to maintain the regular confidence score distribution and thereby preventing extreme miscalibration while undergoing a calibration attack. Since calibration attacks does not flip the original label, For any given classifier, the strongest effect of calibration attacks will be reducing the confidence score on correctly classified "easy" samples while making the model more overconfident on difficult, misclassified samples. This creates a shift in the distribution where for a given high performing classifier the average confidence will drop dramatically while the accuracy remains high, and some misclassified samples will shift to a higher confidence level. In any case, a distribution that was originally skewed towards high confidence scores is now essentially shifted lower. Therein lies the goal of CS, to essentially shift back the distribution by scaling it such that it lies in high confidence space as before. If we assume that already low confidence correctly classified samples will be more affected by a calibration attack than one that is much higher confidence, and if we assume that incorrectly classified samples will have lower confidence then due to the relative inefficiency of the overconfidence attacks they will likely not reach extremely high confidence levels unless the attack is ran for a very large amount of iterations, then the relative ordering between many of the samples is still preserved even if the distribution is shifted, meaning the misclassified samples may still get mapped to the lower end of the confidence scale. The advantage of this method is that it largely does not incur a lot of calibration error even on clean data while being among the most effective and consistent defence methods against calibration attack. In addition, if one wants to do downstream decision making then one can still filter out the bottom $p$ percentage of images with a confidence score. For the number of bins, we mostly choose 3 or 4 as this leads to the smallest error

post attack. We find the scaling factor by iterating through a large range of possible values so that the new desired confidence score for the sample is then achieved within the new confidence range.

## D  CALIBRATION METRIC FORMULATION

Here we forumalate the two calibration metrics that we use in our experiments. As Equation 2 is an idealized representation of miscalibration that is intractable to calculate, approximations have been developed which are grouped into the more common binning-based metrics, and non-binning based metrics.

Expected calibration error (Pakdaman Naeini et al., 2015) is the most widely used calibration metric in research. It is a binning-based metric where confidence scores on the predicted classes are binned into $M$ number evenly spaced bins, which is a hyperparameter that must be carefully chosen. In each bin, the difference between the average confidence score and accuracy of all data points within the bin is calculated, representing the bin-wise calibration error. Afterwards, the weighted sum over the error in each bin constitutes the expectation of the calibration error of the model. The equation for ECE is as follows given $B_m$ are the data points in the $m^{th}$ bin, and $n_m$ is the number of data points in that bin.

$$ECE = \sum_{m=1}^{M} \frac{n_m}{N} |acc(B_m) - conf(B_m)|. \tag{4}$$

ECE can underestimate the levels of miscalibration due to being sensitive to the number of bins (Ovadia et al., 2019) and by having underconfident and overconfident data points overlapping in one bin (Nixon et al., 2020). Kolmogorov-Smirnov Calibration Error (Gupta et al., 2021) is an alternative evaluation metric, that instead of binning, leverages the Kolmogorov-Smirnov statistical test for comparing the equality of two distributions. The error is determined by taking the maximum difference between the cumulative probability distributions of the confidence scores and labels. Specifically, the first step is to sort the predictions according to the confidence score on class $k$, i.e., $\hat{p}_k$, leading to the error being defined as:

$$\begin{aligned} \text{KS error} &= \max_i |h_i - \tilde{h}_i|, \\ \text{where, } h_0 &= \tilde{h}_0 = 0, \\ h_i &= h_{i-1} + \mathbf{1}(y_i = k)/N, \\ \tilde{h}_i &= \tilde{h}_{i-1} + p_k(x_i)/N. \end{aligned} \tag{5}$$

## E  ADVERSARIAL ATTACK DETECTION DETAILS

**Local Intrinsic Dimensionality (LID) (Ma et al., 2018).** This detection method exploits the estimated Local Intrinsic Dimensionality (LID) characteristics across different layers of a model of a set of adversarial examples, which are found to be notably different than that of clean samples or those with added random noise. First, a training set is made up of clean, noisy, and adversarial examples, and a simple classifier (logistic regression) is trained to discriminate between adversarial and non-adversarial samples. For each training minibatch, the input features to the classifier are generated based on the estimated LID across different layers for all of the samples. The two main hyperparameters for this method are batch size and the number of nearest neighbours involved in estimating the LID. We choose a consistent batch size of 100 in line with previous work such as (Harder et al., 2021), and for each case we test the possible nearest neighbors from the following list $\{10, 20, 30, 40, 50, 60, 70, 80, 90\}$ and report the results for the best value, which vary for different datasets and models. We use the implementation of LID from (Lee et al., 2018).

**Mahalanobis Distance (MD) (Lee et al., 2018).** The premise behind this method is to use a set of training samples to fit a class-conditional Gaussian distribution based on the empirical class means and empirical covariance of the training samples. Given a test sample, the Mahalanobis distance with respect to the closest class-conditional distribution is found and taken as the confidence score.

A logistic regression detector is built from this which determines whether a sample is adversarial. The main hyperparameter for this method is the magnitude of the noise used, which we vary between $\{0.0, 0.01, 0.005, 0.002, 0.0014, 0.001, 0.0005\}$ for each case and pick the value that results in the highest detection accuracy. In addition, calculating the mean and covariance is necessary to use the method, which we utilize the respective training set to do for each dataset. We use the implementation of MD from (Lee et al., 2018).

**SpectralDefense (Harder et al., 2021).** This detection method makes use of Fourier spectrum analysis to discriminate between adversarial and clean images. The spectral features from Fourier coefficients, which are computed via two-dimensional discrete Fourier transformation applied to each feature map channel, are found for each image, and a detector based on logistic regression is trained using the Fourier coefficients. The magnitude Fourier spectrum based detector (InputMFS) is the version we use in our experiments.

# F  ADDITIONAL ANALYSIS AND RESULTS

In this section we provide additional results with white-box attacks, more details on the analyses described in Section 4.3, and qualitative analysis of the properties of our attacks, as well as a quantitative analysis under a common real world issue of imbalanced data distributions. Apart from the white-box results, the remaining analyses are conducted using our black-box setup.

## F.1  WHITE BOX VARIATION OF ATTACK

The results for the white-box variation of our attacks can be found in Table 5 on the three datasets and across our two tested models, similar to how we presented our black-box results. For each scenario, we show the ECE, KS error and average confidence. We used 10 attack steps to generate the results for an $\epsilon$ noise value of 0.05.

Much like the SA results, the PGD attack manages to create significant miscalibration compared to before the attack with only a small number of attack steps. The results are less severe than for SA where the level of miscalibration achieved are worse despite the base PGD attack being far more effective at affecting classification accuracy. We believe this is because the modifications that are made to ensure that the calibration attack algorithm does not cause the predicted class to change greatly reduce the effectiveness of PGD as the most effective gradient updates that cause a great swing in the confidence score cannot be used since they are likely to change the predicted class, and instead much less significant updates that do not change the confidence score a great deal serve as the primary noise that gets added to the adversarial images.

## F.2  DETAILED ANALYSIS OF EPSILON AND ITERATIONS

**Epsilon.** The $\epsilon$ parameter plays a major role in adversarial attacks, as it controls how much noise can be added when creating perturbations. Although setting a higher $\epsilon$ value for an attack lets

Table 5: Results of white-box PGD variant of calibration attack.

| ResNet | | | |
|---|---|---|---|
| | **ECE** | **KS** | **Avg. Conf.** |
| *CIFAR-100* (Accuracy: 0.881±0.002) | | | |
| Pre-atk | 0.052±0.006 | 0.035±0.006 | 0.916±0.006 |
| Und-atk | 0.213±0.003 | 0.175±0.007 | 0.747±0.011 |
| Ovr-atk | 0.072±0.003 | 0.07±0.001 | 0.951±0.002 |
| Max-atk | 0.187±0.008 | 0.161±0.007 | 0.746±0.007 |
| Rnd-atk | 0.187±0.016 | 0.156±0.013 | 0.759±0.015 |
| *Caltech-101* (Accuracy: 0.966±0.004) | | | |
| Pre-atk | 0.035±0.002 | 0.031±0.004 | 0.936±0.001 |
| Und-atk | 0.388±0.019 | 0.38±0.022 | 0.599±0.022 |
| Ovr-atk | 0.018±0.003 | 0.019±0.002 | 0.984±0.001 |
| Max-atk | 0.375±0.022 | 0.376±0.022 | 0.591±0.021 |
| Rnd-atk | 0.353±0.019 | 0.352±0.021 | 0.619±0.022 |
| *GTSRB* (Accuracy: 0.972±0) | | | |
| Pre-atk | 0.019±0.006 | 0.008±0.002 | 0.98±0.002 |
| Und-atk | 0.233±0.02 | 0.232±0.016 | 0.752±0.014 |
| Ovr-atk | 0.02±0.002 | 0.019±0.003 | 0.991±0.003 |
| Max-atk | 0.226±0.006 | 0.227±0.007 | 0.75±0.008 |
| Rnd-atk | 0.217±0.014 | 0.218±0.009 | 0.763±0.008 |
| **ViT** | | | |
| *CIFAR-100* (Accuracy: 0.935±0.002) | | | |
| Pre-atk | 0.064±0.006 | 0.054±0.005 | 0.882±0.004 |
| Und-atk | 0.277±0.001 | 0.274±0.004 | 0.671±0.006 |
| Ovr-atk | 0.045±0.003 | 0.017±0.002 | 0.928±0.002 |
| Max-atk | 0.26±0.006 | 0.262±0.007 | 0.675±0.005 |
| Rnd-atk | 0.236±0.013 | 0.239±0.015 | 0.699±0.013 |
| *Caltech-101* (Accuracy: 0.961±0.024) | | | |
| Pre-atk | 0.137±0.059 | 0.136±0.06 | 0.825±0.083 |
| Und-atk | 0.489±0.071 | 0.489±0.071 | 0.472±0.095 |
| Ovr-atk | 0.086±0.045 | 0.082±0.049 | 0.879±0.073 |
| Max-atk | 0.488±0.07 | 0.488±0.07 | 0.473±0.094 |
| Rnd-atk | 0.435±0.048 | 0.435±0.048 | 0.527±0.071 |
| *GTSRB* (Accuracy: 0.947±0.006) | | | |
| Pre-atk | 0.04±0.005 | 0.026±0.017 | 0.922±0.024 |
| Und-atk | 0.321±0.047 | 0.315±0.045 | 0.641±0.052 |
| Ovr-atk | 0.037±0.013 | 0.02±0.011 | 0.936±0.024 |
| Max-atk | 0.302±0.037 | 0.302±0.036 | 0.645±0.043 |
| Rnd-atk | 0.292±0.03 | 0.293±0.029 | 0.657±0.037 |

it easier and more efficient for the algorithm to create adversarial examples, it potentially cause the visual changes to images more perceptible, so a small $\epsilon$ is preferable while still being able to produce good adversarial examples. In the case of calibration attack, there is no need to go as far as flipping a label, so lower $\epsilon$-bounds have the potential to create some miscalibration. To provide further details on our results in Figure 2, for the leftmost figure as mentioned previous we tested on CIFAR-100 using ResNet-50. The five different $\epsilon$ values we use are (0.005, 0.01, 0.05, 0.1, 0.25) after being attacked using all four of the attacks with the other settings the same as in Appendix B, with the results averaged over three models. In addition to the miscalibration being strong for most of the attacks at low $\epsilon$ values, we can see that maximum miscalibration attack consistently outperforms the rest across the different values. The underconfidence attack does not have much change with higher $\epsilon$, but it is largely because the models has already almost reached the peak level of attacking the confidence with low epsilon values, and as such does not have a large effect on ECE. As middle figure largely displays the same trends as ResNet, revealing that the results are not architecture dependant. The rightmost figure uses ResNet and goes over the same $\epsilon$ values as before, except the maximum miscalibration attack is run over both Caltech-101 and GTSRB models. Again the trends are similar, although the increase in ECE is not as severe as for CIFAR-100.

**Iterations.** Expanding on the results in the rightmost figure in Figure 2, the number of iterations of the maximum miscalibration attack is varied from 100, 500, 1000, to 5000, whilst attack both ViT and ResNet models trained and tested on CIFAR-100 with the same settings as in Appendix B. We note how the ECE begins to saturate at close to 500 iterations, after which the benefits of running the attack longer are minor, though even at 100 iterations the ResNet model becomes heavily miscalibrated despite the standard $\epsilon$ value of 0.05 being used. In our tests showing the effectiveness of the original SA versus its calibration attack version, seen in Figure 4 we specifically compare over accuracy and ECE between the maximum miscalibration attack and the regular untargeted Square Attack across the four aforementioned iteration values for both ResNet and ViT on CIFAR-100. As expected, Square Attack greatly reduces the accuracy even with a small number of iterations. Nevertheless, in terms of ECE, the calibration

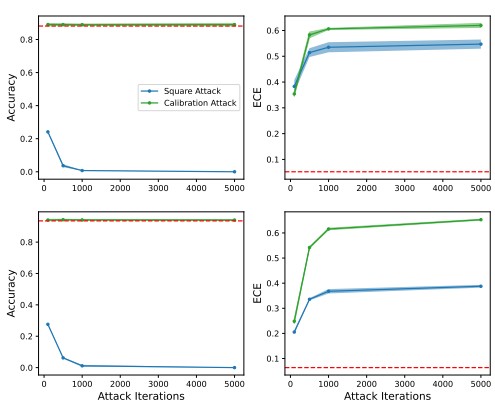

Figure 4: The contrast between the effects on accuracy and ECE between the original version of the Square Attack algorithm and the maximum variation of the calibration attack algorithm at 1000 attack iterations. (Top) ResNet-50 results. (Bottom) ViT results.

attacks consistently produce higher amounts of miscalibration compared to the original Square Attack across the different iteration amounts.

### F.3 t-SNE VISUALIZATIONS

To help visualize the effect of each of the attack types in latent space and to confirm they are having the expected effects, we run a t-SNE analysis (van der Maaten & Hinton, 2008) on the representations of ResNet-50 right before the classification layer. The datasets we use throughout this study, with their large number of classes, are not ideal for visualization purposes. Instead, we create a binary subset using CIFAR-100 by taking all of the images from two arbitrary classes, bicycles and trains. We create a separate training set and test set to perform this procedure independently, and fine-tune a ResNet-50 model on the training set. The specific details are similar to those described in Section C for CIFAR-100 ResNet. We train the model for 5 epochs with a learning rate of 0.005. The attack settings are the same as in Section B for the $l_\infty$ version, and we only run the attacks for 500 iterations. We run the t-SNE analysis on a balanced slice of 200 images from the new test set for easy visualization purposes, before and after all of the different attacks. The model achieves 95% accuracy on the full test set. Figure 5 shows the graphs. It can be seen the effect on the representations for the adversarially attacked data is as expected. The overconfidence attack causes the representations for both class predictions, even incorrect ones, to be split apart as much as possible, while the underconfidence attack causes a more jumbled representation between the two classes

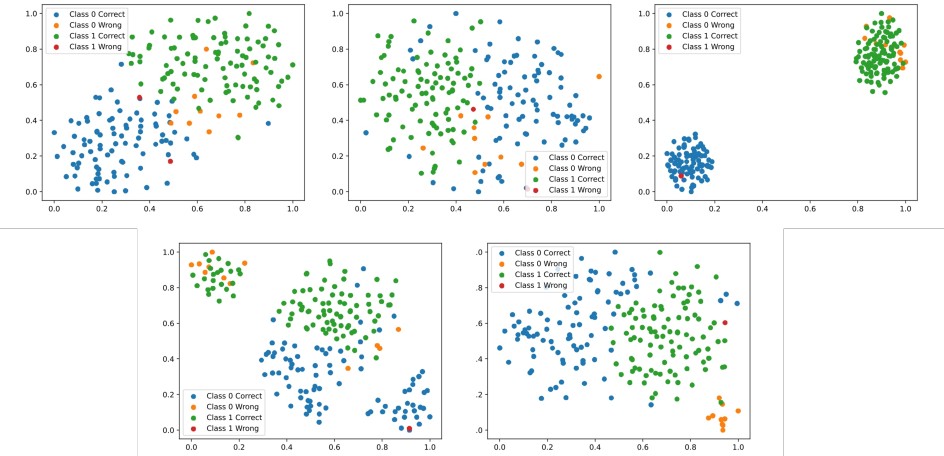

Figure 5: t-SNE visualization of the effect of different forms of calibration attacks on a ResNet model trained and tested on a binary subset from CIFAR-100, with the test set (consisting of 200 data points) results being displayed. In the order from top left to bottom right, the plots for the pre-attack (vanilla model), and the underconfidence, overconfidence, random, and maximum variations of the attacks can be seen.

with most falling closely to the decision boundary. The maximum miscalibration attack has a similar effect to the underconfidence attack, except the misclassified images are pushed far away from the decision boundary to make it appear as if the model is more confident in its decisions. Lastly, the random attack causes two distinct random clusters for each prediction type to form, as random data points are pushed to be more overconfident or more underconfident than they originally were. With these results, we can see visually confirm that the attacks possess their intended behaviour.

## F.4 IMBALANCE RATIO

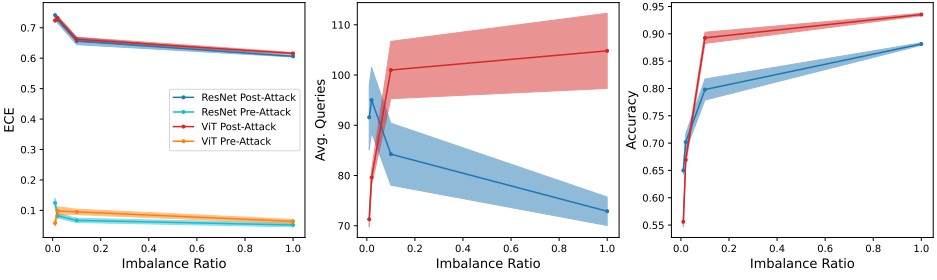

Figure 6: Graphs comparing the vulnerability of ResNet and ViT models trained with different imbalance ratios on CIFAR-100 to the maximum miscalibration attack at 1000 iterations and an $\epsilon$ of 0.05, and their corresponding overall trends in average queries and accuracy.

Dataset imbalance has a profound effect on how a model learns and how well it performs, with detrimental effects occurring when imbalance ratios are very high. With how common imbalanced data distributions are in real world scenarios, we believe it is worth studying the influence of imbalance ratio and its relationship with robustness against calibration attacks as an additional point of analysis. We choose CIFAR-100 as our primary dataset for this analysis, and we follow the procedures in Tang et al. (2020) and Cao et al. (2019) to create training sets with long-tail imbalance. This is a form of imbalance where the sample sizes in the classes follow an exponential decay. We use the variable $\rho$ to denote the ratio between sample sizes of the class with the smallest sample size, and those of the one chosen to be the biggest. We create training sets with $\rho$ values of 0.01, 0.02, and 0.01 (for 1:100, 1:50 and 1:10 ratios of smallest to biggest class). We then train 3 ResNet-50 and 3

ViT models on each imbalanced set. The training details are again the same as those described in the general settings Section C, although the training times are different. 15 epochs is used to training the 1:100 ratio models, while 10 epochs is used for the rest. We subject the models to the maximum miscalibration attack using the same settings as in Section B for CIFAR-100 (test data is balanced), and calculate the resulting average and deviation of the pre and post attack ECE, average number of queries, and accuracy. The graphs displaying the results can be seen in Figure 6. Unsurprisingly, the higher the imbalance ratio, the lower the accuracy is on the balanced data. In terms of robustness, the more balanced the data the more resistant it is against getting miscalibrated from the attacks, for both the ResNet and ViT architectures. This is similar to the trends in the inherent miscalibration present before the attacks, although the calibration differences between the different ratio models are not as severe, and ViT at the 1:500 imbalance ratio is the best calibrated beforehand but becomes the worst after the attack. The trend in the number of queries it takes for a successful attack is reversed for ResNet and ViT, with ViT requiring more queries the more balanced the data is, while ResNet is vice-versa. Overall, dataset imbalance does not create favourable conditions for robustness, though the use of imbalance data techniques could potentially remedy some of these issues.

### F.5   GRADCAM VISUALIZATION DETAILS

Given the effectiveness of the attacks at leading a model to produce highly miscalibrated outputs, for both base styles of attacks, we endeavour to explore whether they also lead to any changes in where the model focuses on in an image when making its decision, and especially with novel overconfidence attack. Knowing this can lead to further insights as to how models are affected by various forms of the attacks. To accomplish this analysis, we use Grad-CAM (Selvaraju et al., 2017), a popular visualization method that produces a coarse localization map highlighting the most important regions in an image that the model uses when making its prediction by making use of the gradients from the final convolutional layer (or a specific layer of choice) of a network. We apply Grad-CAM to our ResNet-50 models fine-tuned on Caltech-101 to images from the Caltech-101 test set before and after the underconfidence and overconfidence attacks at the standard attack settings used in Section B and using the GradCAM implementation of Gildenblat & contributors (2021). Since the method calculates relative to a specific class, we do so in-terms of the predicted class. Figure 3 shows the results with some representative images. We specifically choose images where the attacks led to large change in predicted confidence (at least 10%). On the whole, we have observed that the coarse localization maps have minimal to no noticeable changes after the adversarial images are produced, especially in the case of the overconfidence attacked images. This leads us to believe the primary mechanism of the attacks changing the model confidence is in the final classification layer as opposed to the convolutional layers. This analysis also shows that it might be difficult to identify these attacks are occurring based on these types of gradient visualization methods.

## F.6 $L_2$ RESULTS

Table 6: Results of the $l_2$ calibration attacks for the three different datasets.

**ResNet**

|  | avg #q | median #q | ECE | KS | Avg. Conf |
|---|---|---|---|---|---|
| *CIFAR-100* | | | | | |
| **Accuracy:** 0.881±0.002 | | | | | |
| Pre-Attack | - | - | 0.052±0.006 | 0.035±0.006 | 0.916±0.006 |
| Underconf. Atk. | 182.7±13.0 | 94.0±11.0 | 0.399±0.012 | 0.356±0.010 | 0.566±0.008 |
| Overconf. Atk. | 44.6±3.3 | 1.0±0.0 | 0.129±0.007 | 0.129±0.007 | 0.995±0.000 |
| Maximum Atk. | 137.3±5.8 | 92.7±10.6 | 0.496±0.001 | 0.391±0.002 | 0.604±0.003 |
| Random Atk. | 125.2±7.3 | 99.7±4.9 | 0.431±0.016 | 0.350±0.012 | 0.614±0.011 |
| *Caltech-101* | | | | | |
| **Accuracy:** 0.966±0.004 | | | | | |
| Pre-Attack | - | - | 0.035±0.003 | 0.031±0.004 | 0.936±0.001 |
| Underconf. Atk. | 293.5±14.9 | 195.0±61.7 | 0.156±0.002 | 0.157±0.003 | 0.810±0.002 |
| Overconf. Atk. | 60.9±4.4 | 1.0±0.0 | 0.019±0.004 | 0.017±0.006 | 0.982±0.002 |
| Maximum Atk. | 40.8±1.8 | 227.2±91.9 | 0.143±0.006 | 0.140±0.005 | 0.836±0.005 |
| Random Atk. | 33.5±5.3 | 205.0±38.3 | 0.120±0.009 | 0.121±0.008 | 0.848±0.008 |
| *GTSRB* | | | | | |
| **Accuracy:** 0.972±0.000 | | | | | |
| Pre-Attack | - | - | 0.019±0.006 | 0.008±0.002 | 0.980±0.002 |
| Underconf. Atk. | 291.5±22.4 | 196.7±16.6 | 0.190±0.032 | 0.187±0.029 | 0.793±0.030 |
| Overconf. Atk. | 19.5±3.3 | 1.0±0.0 | 0.022±0.002 | 0.022±0.002 | 0.997±0.000 |
| Maximum Atk. | 91.4±21.1 | 142.8±44.8 | 0.239±0.038 | 0.225±0.034 | 0.771±0.035 |
| Random Atk. | 97.5±16.8 | 211.0±41.1 | 0.200±0.014 | 0.191±0.011 | 0.794±0.011 |

**ViT**

|  | avg #q | median #q | ECE | KS | Avg. Conf |
|---|---|---|---|---|---|
| *CIFAR-100* | | | | | |
| **Accuracy:** 0.935±0.002 | | | | | |
| Pre-Attack | - | - | 0.064±0.006 | 0.054±0.005 | 0.882±0.004 |
| Underconf. Atk. | 199.6±7.1 | 111.2±12.5 | 0.383±0.014 | 0.382±0.013 | 0.555±0.011 |
| Overconf. Atk. | 681.3±408.4 | 681.3±408.4 | 0.022±0.002 | 0.021±0.003 | 0.958±0.003 |
| Maximum Atk. | 111.9±7.4 | 131.5±17.7 | 0.405±0.010 | 0.383±0.010 | 0.590±0.007 |
| Random Atk. | 108.2±6.9 | 137.8±17.0 | 0.343±0.010 | 0.334±0.007 | 0.614±0.004 |
| *Caltech-101* | | | | | |
| **Accuracy:** 0.961±0.024 | | | | | |
| Pre-Attack | - | - | 0.137±0.059 | 0.136±0.060 | 0.825±0.083 |
| Underconf. Atk. | 258.7±47.2 | 207.8±64.0 | 0.233±0.057 | 0.233±0.057 | 0.729±0.081 |
| Overconf. Atk. | 23.2±15.0 | 1.0±0.0 | 0.100±0.048 | 0.100±0.048 | 0.859±0.073 |
| Maximum Atk. | 31.5±2.2 | 236.5±52.0 | 0.224±0.058 | 0.224±0.058 | 0.740±0.080 |
| Random Atk. | 21.9±8.1 | 293.8±24.5 | 0.196±0.038 | 0.196±0.038 | 0.764±0.064 |
| *GTSRB* | | | | | |
| **Accuracy:** 0.947±0.006 | | | | | |
| Pre-Attack | - | - | 0.040±0.005 | 0.026±0.017 | 0.922±0.024 |
| Underconf. Atk. | 258.3±27.8 | 169.7±31.5 | 0.261±0.012 | 0.262±0.011 | 0.686±0.016 |
| Overconf. Atk. | 70.2±31.3 | 1.0±0.0 | 0.030±0.005 | 0.024±0.012 | 0.968±0.016 |
| Maximum Atk. | 99.6±10.9 | 210.5±33.0 | 0.274±0.037 | 0.257±0.038 | 0.718±0.044 |
| Random Atk. | 94.7±11.2 | 213.8±59.9 | 0.245±0.020 | 0.241±0.016 | 0.714±0.007 |

