# OpenReview forum: "Calibration Attack: A Framework For Adversarial Attacks Targeting Calibration"
_ICLR.cc/2024/Conference — Submitted to ICLR 2024_

### Official Review · Reviewer_BDLx · 2023-10-19

**Soundness:** 4 excellent
**Presentation:** 2 fair
**Contribution:** 3 good
**Rating:** 5
**Confidence:** 3

**Summary:**

This paper introduces calibration attacks, a new class of adversarial attacks that aim to miscalibrate the confidence scores of models without changing their accuracy. The authors propose four types of calibration attacks and demonstrate their effectiveness against image classifiers like ResNet and ViT across datasets. The attacks are difficult to detect using common adversarial defense techniques. Analyses show the attacks modify model representations and confidences as expected while minimally impacting gradient-based visualizations. Existing defense methods like temperature scaling and adversarial training provide limited robustness against calibration attacks. The authors propose two new tailored defenses and analyze model vulnerabilities, highlighting the need for further research into mitigating this dangerous new attack vector which could seriously impact reliability if deployed against real-world systems.

**Strengths:**

The idea of attacking the calibration of ML models is novel, and the work (analyses, discussion, etc.) is solid. Particularly, the authors conducted extensive experiments on adversarial attacks and defenses.

**Weaknesses:**

See **Questions** part.

**Questions:**

1. Questions about Figure 1.
As mentioned in its caption, red bars represent accuracy and blue bars represent confidence. However, the vertical axis in your image is also labeled as accuracy. Could you explain this in detail? Besides, I noticed that Figure 1 is in PNG format. A vector graphic format might be better.

2. How to calculate the average confidence?
In my opinion, letting "average confidence = the product of the sample proportions in each bin" would provide a reasonable explanation according to your setting. However, in Table 1, I think the average confidence is not calculated in this way. Otherwise, we could have Accuracy = ECE +/- average confidence.

3. Relation with previous works
I list two more related works on robust calibration. It would be perfect if you could compare your work with theirs.
[1] Tang Y C, Chen P Y, Ho T Y. Neural Clamping: Joint Input Perturbation and Temperature Scaling for Neural Network Calibration, arXiv:2209.11604.
[2] Yu Y, Bates S, Ma Y, et al. Robust calibration with multi-domain temperature scaling, in NeurIPS 2022.

4. The implementation of adversarial attack/defense algorithms.
I found it very hard to parse the results of the experiments. I wonder if the authors could provide the corresponding code to the results in Table 1 (or at least a demonstration) so that I can reproduce and check the results.

I will be very happy to reconsider my rating if the authors could address my concerns.

**Details Of Ethics Concerns:**

I have no ethical concerns for this paper.

---

> ### Author Response · Authors · 2023-11-17
>
> Q1: The x-axis represents the confidence range, and the y-axis is the accuracy in that range of confidence scores.
>
> Q2: Average confidence in the tables is referring to the averaged predicted confidence score for all of the tested data. In the context of the ECE calculation it is the same but relative to a specific bin.
>
> Q3: Although it would be interesting to test these additional baselines, we have already tried many baseline defense methods and we cannot test every possible method of this type. We believe the current methods we tested are a good overview of the general performance.
>
> Q4: The code for the results in table 1 and the rest of the results will be provided later as we will put the full code on GitHub. For now, are there any specific questions about the implementation as we cover our algorithm and training details in detail in the paper?

---

### Official Review · Reviewer_YLZS · 2023-10-28

**Soundness:** 3 good
**Presentation:** 1 poor
**Contribution:** 1 poor
**Rating:** 1
**Confidence:** 3

**Summary:**

This paper proposes a test-time attack that manipulates a classifier’s confidence scores without changing the classifier’s prediction. Two types of manipulation are considered: (1) increasing the classifier’s confidence for the predicted class and (2) decreasing the margin between the confidence of the predicted and runner-up class. These manipulations can be applied to inputs randomly or to maximize miscalibration. The attacks are carried out using a variant of the square attack (a gradient-free algorithm) or projected gradient descent and are shown to be effective empirically on standard image datasets. The paper also studies the effectiveness of the attack under four calibration methods (e.g., temperature scaling, splines) and under defenses such as adversarial training.

**Strengths:**

The paper is comprehensive in that it covers several variants of the proposed attack (black-box, white-box, different confidence manipulations) while also exploring potential defenses. The empirical results are extensive: there is a good selection of datasets, models, and evaluation metrics.

I appreciate that the paper studies a non-conventional threat model. A benefit of the proposed attack is that it has the potential to cause harm with a smaller perturbation strength compared to adversarial examples that cause misclassification. For that reason, it may also be more less susceptible to detection, particular if the confidence scores are manipulated randomly.

**Weaknesses:**

1. My primary concern with the paper is around originality and its failure to cite prior work. The paper claims to be the first to study attacks on confidence scores, however a very similar test-time attack was proposed by Galil & El-Yaniv (2021). More recently a training-time attack on confidence scores was proposed by Zeng et al. (2023) (which first appeared on arXiv in 2022). It’s unfortunate these papers are not cited. There is also closely related work on certifying confidence scores by Kumar et al. (2020) and Emde et al. (2023) which ought to be cited.

2. In order to better assess originality, I have compared this paper with Galil & El-Yaniv (2021). As far as I can tell, the differences between the attacks are minor:
    - Galil & El-Yaniv focus on reducing the confidence of the predicted class, whereas this paper also considers increasing the confidence.
    - Galil & El-Yaniv’s attack algorithm is FGSM-based, whereas this paper uses the Square attack algorithm and PGD.
    - Galil & El-Yaniv maximize/minimize the confidence of the predicted class directly, whereas this paper also considers the margin.
Apart from attacks, this paper also contributes some insights on defences, which is not something that Galil & El-Yaniv cover. However, overall I don’t believe this paper’s contributions are sufficiently original/significant, at least in its current state.

3. Regarding the presentation of Section 5: I found it confusing that the calibration methods (TS, Splines, DCA, SAM) are presented alongside the defenses (AAA, AT, CA AT, CS). I think there is a risk that some readers may misinterpret the calibration methods as defenses, even though they are not designed to defend against attacks.

4. A missing baseline defense: Kumar et al. (2020) propose Gaussian randomized smoothing as a method with certified guarantees on the confidence scores. While their guarantees may not be in perfect alignment with the $\ell_\infty$ attacks proposed in this paper, I think it’s an important baseline to include, given it’s designed to produce more robust confidence scores.

5. Clarification of threat model: in order to conduct the _maximum miscalibration attack_ I believe the attacker needs to know the ground truth for each input, so they can perturb the confidence in the direction that causes maximum miscalibration. I wonder whether it is realistic to assume the attacker knows the ground truth for all inputs they want to attack. If they were able to obtain the ground truth cheaply, then it may suggest that the classification problem is not so difficult.

**References**

- Kumar et al., “Certifying Confidence via Randomized Smoothing,” NeurIPS 2020. https://proceedings.neurips.cc/paper_files/paper/2020/file/37aa5dfc44dddd0d19d4311e2c7a0240-Paper.pdf

- Galil and El-Yaniv, “Disrupting Deep Uncertainty Estimation Without Harming Accuracy,” NeurIPS 2021. https://proceedings.neurips.cc/paper/2021/file/b1b20d09041289e6c3fbb81850c5da54-Paper.pdf

- Zeng et al., “Manipulating Out-Domain Uncertainty Estimation in Deep Neural Networks via Targeted Clean-Label Poisoning,” CIKM’23 https://dl.acm.org/doi/abs/10.1145/3583780.3614957

- Emde et al., “Certified Calibration: Bounding Worst-Case Calibration under Adversarial Attacks,”
AdvML-Frontiers'23 Workshop. https://openreview.net/forum?id=sj5K9jtrdm

**Questions:**

1. Given the existence of prior work on calibration attacks, you could consider focusing more on the defense side (where there has been less work) or on variations of the threat model.

2. For underconfidence attacks, the attack minimizes the margin between the scores of the predicted and runner up classes. Have you considered optimizing the scores of the other classes as well? For instance, one could imagine trying to perturb the scores to be close to uniform by maximizing the entropy of the scores.

3. It seems the defenses are tested under the maximum miscalibration attack. I wonder how the results would differ for the random miscalibration attack. In particular, I wonder whether the compression scaling defense would be effective in that setting, since it seems to make strong assumptions on the way in which scores are perturbed.

4. Is it possible to transfer these attacks to other classifiers? I expect that transfer may risk harming accuracy, especially if the attacked points are moved closer to the decision boundary. Perhaps over-confidence attacks are more reliable for transfer?

---

> ### Author Response · Authors · 2023-11-17
>
> Weaknesses 1 and 2: We greatly appreciate making us aware of these other previous works that are in a similar vein to our paper, and we will acknowledge and reference them in our revision. Nevertheless, we still believe that our contributions over these works are substantial, as none of the prior works comprehensively study the range of the 4 attacks we introduce and cover in the context of calibration rather than uncertainty estimation, nor do they systematically investigate how well victim models would remain well calibrated under such attacks, including under a range of different calibration methods and adversarial defense methods, including a few novel ones like CAAT and CS. Unlike previous works, we examine in detail the effects of attacks targeting overconfidence as well, and evaluate the detectability relative to standard attacks both in terms of detectability methods and visualization approaches like Grad CAM.
>
> Weakness 3: Indeed, they are not traditional defenses, but in the context of our work we are trying to examine whether by trying to better calibrate a model through traditional calibration methods leads to any sort of robustness against calibration attacks.
>
> Weakness 4: Although this baseline would be an interesting point of comparison, we do already test methods meant for robust confidence scores like AAA. In addition, the L2 bounded norm used as the bases in the Kumar et al. (2020) paper means it would be difficult to use it as a comparison given the primary focus of our analysis and results in terms of defenses is in terms of ℓ∞
>
> Weakness 5: That is true that it requires the attack to know the ground truth, but for most conventional adversarial attack papers it is already assumed the attack knows the correctly classified samples and attacks them to result in a misclassification, hence we do not see how this is all that different from that. In addition, the maximum attack is meant to be the strongest case. One can still utilize all of the other attacks, and in many cases the random attack performs quite similarly to the maximum miscalibration attack, and it requires no knowledge about the labels, so the harm from calibration attacks is still great.
>
> Q1: We believe our experiments and analysis are novel, particularly for the overconfidence attack threat model which has not been studied before in previous work.
>
> Q2: That is an interesting notion. We focus on the predicted class as it is more query efficient than trying to target every single possible class, especially in the black box setting, especially when the number of classes is very high (like 100) and many have very low probabilities which would be difficult to raise to yield a uniform distribution given the relative inefficiency of the overconfidence attack. (What’s the application of the suggested experiments)
>
> Q3: We chose the maximum because it is the strongest form of attack and hence offers the best stress test to the defense methods. While its true that CS targets being effective against the maximum attack, it is primarily designed to be against it to avoid worse case miscalibration through the attacks. There is no perfect method to resist all forms of calibration attacks yet, which is why we open up the issue to the community to find methods that work across all forms of attacks.
>
> Q4: Even the overconfidence attacks can affect the accuracy as we have noted in practice, particularly on samples close to the decision boundary, so we must reject a certain number of updates because of this, so we believe attack transfer might be difficult, at least with maintaining the goal of leaving the accuracy unaltered.

---

> > ### Comment · Reviewer_YLZS · 2023-11-18
> >
> > Thank you for your detailed response.
> >
> > > Weaknesses 1 and 2: We greatly appreciate making us aware of these other previous works that are in a similar vein to our paper...
> >
> > I agree that there are novel aspects in this paper, particularly the results on defenses. However, I think the paper requires substantial editing to reposition/recalibrate the contributions and acknowledge the existence of prior work. This would require another round of review in my opinion, which is unfortunately not possible at ICLR.
> >
> > > Weakness 5: That is true that it requires the attack to know the ground truth, but for most conventional adversarial attack papers it is already assumed the attack knows the correctly classified samples and attacks them to result in a misclassification, hence we do not see how this is all that different from that. In addition, the maximum attack is meant to be the strongest case. One can still utilize all of the other attacks, and in many cases the random attack performs quite similarly to the maximum miscalibration attack, and it requires no knowledge about the labels, so the harm from calibration attacks is still great.
> >
> > I agree that it is common to assume knowledge of ground truth for conventional evasion attacks. This seems practical if the attacker is targeting a small number of instances, but less practical if the attacker is targeting a large number of instances. I take your point about the random attack, which doesn't require knowledge of ground truth.

---

### Official Review · Reviewer_PTEC · 2023-10-30

**Soundness:** 3 good
**Presentation:** 3 good
**Contribution:** 2 fair
**Rating:** 5
**Confidence:** 3

**Summary:**

The authors proposed four types of adversarial attacks targeting calibration specifically, which maximizes the error in model’s prediction score without alternating the predicted label/accuracy. Building on top of some existing successful adversarial attack technique (e.g. SA), the authors achieve miscalibration through lowering the prediction on correctly classified cases and increasing the prediction on incorrect cases. It is shown to be effective on many models/datasets even in the presence of popular calibration methods. Authors also discussed the parameter choices and how they affect the performance of the attack.  In addition, the authors proposed new calibration methods that are capable of defending the attacks proposed in this paper.

**Strengths:**

- The subject of this research seems novel. It is distinctive from most types of adversarial attack studies that aims to increase misclassification and lower the predictive accuracy. Instead, it has the constraint of not affecting this most notable performance metric while maximizing the error in the confidence level, which is often overlooked.
- The experiments have good coverage on different cases and sufficiently demonstrated the authors' conclusion. The discussion is also thorough about the design choices in both the attack algorithm and the defending methods.
- The paper is clearly written and easy to follow.

**Weaknesses:**

- The authors' motivation for calibration attack is that the prediction score instead of the classified label is directly used in downstream tasks. However, most of the experiment results are presented in terms of the calibration error, but did not explore further on its effect on the downstream. Without a concrete example, it is hard to assess the significance on the implication of this error on confidence score.
- While the idea of attack on calibration is new as far as I know, the attack mechanism is largely based on the existing methods and does not seem to have significant novelty in itself.
- In terms of the defense method, it is unclear whether it only addresses the attacks proposed in the paper or has more general effectiveness. Based on the pre-attack result in table 4, it doesn't seem to be superior than other existing calibration methods in terms of fixing general miscalibration, which gives me the concern that it might be only effective towards these specific attacks.

**Questions:**

- The attack algorithm is independent between samples, but is it possible to inferred information about the model on its tendency to be over/under-confident and reduces the query sizes?
- Is it still possible to perform the attack if only the predicted class label can be queried?
- Is it possible to attack the training process and have the model be over/under-confident on unaltered samples?

---

> ### Author Response · Authors · 2023-11-17
>
> Motivation: The effect on average classification confidence can be seen, including a t-sne diagram in the appendix showing the effects of the attacks on the decision boundary. Calibration error is also the primary means of determining how suitable a model’s predicted confidence scores are. We can refer to papers such as Confidence Estimation and Deletion Prediction Using Bidirectional Recurrent Neural Networks (Ragni et al.) where in an ASR system downstream confidence scores are used for flagging up words that need further review for not wasting resources annotating data that is too easy, in which case edge cases are identified through confidence scores. Both of these require low calibration error for example. In the both cases, overconfidence attacked data will make it impossible to do these tasks. We can add these discussions in our revision.
>
> Novelty: Please refer to rebuttal to Reviewer PTEC
>
> Q1: That is an interesting question. We thought about this previously, but determined that it does not make that big of a difference targeting a model’s existing weaknesses by estimating its distribution because the max and random attacks tend to be more effective in the long run. One can implement targeting the specific calibration distribution this naturally by choosing the corresponding one of the attack types if one wishes (e.g., if average predicted confidence is far higher than accuracy then use overconfidence attack). The 4 different attacks we create offer a lot of flexibility in terms of how you would want to attack a model, and on which. Many derived attacks can be created, if same one determines that at a confidence level of 70% the model is very inaccurate, hence we should use the underconfidence or overconfidence attack to place more incorrectly classified samples into that range. This could be more efficient if one is using very small query sizes, but in the long run the maximum attack will be more effective, and the random attack will completely result in scrambled confidence scores.
>
> Q2: Based on the current implementation, it is unlikely, though you could pick a black box attack that only makes use of predicted class labels and run the maximum attack on the samples based only the class labels and we believe the effect should be the same, although with less efficiency.
>
> Q3: This is interesting but out of scope for this paper as we assume that classifiers are already trained and this scenario would be more of a data poisoning attack. In addition, existing models tend to lean towards overconfidence particularly when training on cross-entropy loss and that this is an inherent learning bias. We know that by feeding calibration attacked samples the inherent calibration of the model (on clean data) is made worse as it does with all of the adversarial training methods, but a sort of data poising approach we do not think it is as viable a scenario as we primarily want to introduce the issue in its most likely form.

---

### Official Review · Reviewer_LAwv · 2023-11-01

**Soundness:** 3 good
**Presentation:** 3 good
**Contribution:** 2 fair
**Rating:** 5
**Confidence:** 4

**Summary:**

The paper proposes a new type of problem to attack the calibration of a DNN without misleading its prediction results. Four attack goals are set: underconfidence attacks, overconfidence attacks, maximum miscalibration attacks, and random confidence attacks. The authors achieve the goals by designing new attack loss, and using existing white-box and black-box attack algorithms. Comprehensive experiments validate the effectiveness of the method.

**Strengths:**

1. It is good to see a proposal for a new attack problem.
2. The authors explore different attack scenarios in the new problem.
3. The conducted experiments are extremely extensive. I appreciate the comprehensive evaluation of various white-box, black-box attacks, as well as the defenses.
4. The paper is well-written and easy to follow, and it gives a good survey of existing work.

**Weaknesses:**

1. The technical contribution is not strong. The modification from misleading the prediction to misleading the calibration is very straightforward. Because misleading the prediction is achieved by controlling the logits, making the ground-truth logits lower than other logits. And certainly, it is easy to manipulate the logits to be any distribution, hurting the calibration.

2. It would benefit more readers if the authors put their emphasis on the significance of calibration attack, instead of the specific method. Since the paper is proposing a new problem, which is brave, the most important thing would be claiming that it is a worthwhile thing. I am not convinced by the only illustration of autonomous driving in the introduction. The design of each method looks too complicated and distractive.

3. It may be a more concise and clear way to present Algorithm 1 and the two defense methods.

**Questions:**

Response to rebuttal: Thanks for the rebuttal. I agree that this is a decent work, but the contribution is not significant enough for ICLR, as also mentioned by Reviewer YLZS. The efforts are far from sufficient to convince the community that the new problem is significant.

---

> ### Author Response · Authors · 2023-11-17
>
> It is true that the extension is not complicated, but many papers present techniques or methods that are simple modifications or variations of previous one, like the temperature scaling technique in the famous Guo et al. 2017 calibration paper, which was a simple modification of the previous Platt Scaling method, but provide strong results or present an interesting empirical analysis. We believe our paper fits into this category as no one has looked at adversarial attacks from the standpoint of attacking calibration without affecting the accuracy, and particularly examined aspects like efficiency, detection rate, defense method effectiveness etc. This is simply not an issue that the community is aware of, and without this paper vulnerable systems may not have the insights to see how these kinds of attacks can occur and comprise their performance. In particular studying attacking through overconfidence is not something that was really featured in previous works, and we believe that the results are interesting, including the insight of how it is more difficult to attack overconfidence than underconfidence.
> The autonomous driving example is only one such example that we put in because of the GPTSRB dataset. We can add and discuss an example in the medical domain where a doctor is using a model to determine whether x-ray scans contain cancer. In this case by attacking the scans and making the model miscalibrated without compromising the accuracy, edge cases that may or may not contain cancer can be made to be overconfident, meaning it is easier for the domain expert to ignore them. Vice versa, easy cases that contain no cancer can be made significantly underconfident, flooding the domain expert (doctor) with a lot of unnecessary work double checking scans. We will update this in our revision to make it clearer.

---

### Meta-Review · Area_Chair_RrJs · 2023-12-06

**Metareview:**

This paper studies test-time attacks that can manipulate model calibration levels without changing the classifier’s prediction. The main techniques are built based on existing successful adversarial attack methods. The paper covers 4 attack methods and validates the effectiveness of the attack under 4 calibration methods (e.g., temperature scaling, splines) and under defenses such as adversarial training.

Strength: all reviewers agree that this problem is important and meaningful. The coverage in attack and calibration techniques is good and the experiments are extensive.

Weakness: the major concerns after the rebuttal are the novelty and the comparison with prior work. When the attack technique is not novel enough, it would be nice if the implication of attacked calibration can be more elaborated, including the downstream tasks. Also, with a large body of previous work in the adversarial attack and the calibration domain, it would be helpful to recalibrate the relation with related work.

**Justification For Why Not Higher Score:**

Several concerns still exist after the rebuttal.

**Justification For Why Not Lower Score:**

N/A

---

### Decision · Program_Chairs · 2024-01-16

Reject